# Zonation of the *Vitis vinifera* microbiome in Vino Nobile di Montepulciano PDO production area
Giorgia Palladino[1,2], Enrico Nanetti[1], Daniel Scicchitano [1,2], Nicolò Cinti[1,2], Lucia Foresto [1,2], Alice Cozzi[1], Antonio Gonzalez Vara Rodriguez[3], Nicolò Interino[4], Jessica Fiori[4,5], Silvia Turroni [1], Marco Candela [1,2] & Simone Rampelli [1,2] ✉

The microbial dimension of the terroir is crucial for wine quality, as microbiomes contribute to plant biofertilization, stress tolerance and pathogen suppression. While microbial terroir can act as a biological signature at large scale, data for local contexts is lacking, hindering the characterization of regional microbial diversity in vineyards. Here, we define the microbial terroir of vineyards across the 12 sub-areas (Additional Geographic Units -AGUs) of the "Consorzio del Vino Nobile di Montepulciano DOCG" PDO area (Italy), a world-renowned wine-producing region. Rhizospheres of *Vitis vinifera* cultivar Sangiovese and soil samples were collected throughout the 2022 viticultural season and analyzed through an integrated metabarcoding/shotgun metagenomic approach, targeting bacteria and fungi. Wine metabolomics was also perfomed, projecting compositional and functional variations of the microbial terroir at the AGUs level into a corresponding variation in the product metabolic profile. Our findings reveal a unique taxonomic configuration of the Vino Nobile di Montepulciano terroir compared to other vineyards, with microbiomes being "AGU-specific" in taxonomic abundances and plant growth-promoting functions, confirming the potential relevance of characterizing and preserving the microbial terroir to safeguard high-quality traditional wines.

It is culturally common to associate wine with the place of production, with specific and recognizable characteristics, so much so that the place of origin is one of the main factors guiding wine purchase decisions[1]. The uniqueness of the relationship between wine and its territory of origin is defined by the concept of terroir, which includes local pedoclimatic, biotic and abiotic factors, combined with traditional agricultural practices, to explain the distinctive regional characteristics of the product[2] (International Organization of Vine and Wine, Definition of vitivinicultural "terroir"— https://www.oiv.int/public/medias/379/viti-2010-1-en.pdf). Today, the concept of wine terroir has spread throughout the world and is regulated by wine-producing countries through the legal definition of appellations of origin, such as the Protected Designation of Origin (PDO) in Europe. In Italy, wines made from identical grape cultivars but grown in different PDO areas with similar yields, are recognized as different products with different organoleptic characteristics[3–5]. Therefore, much is attributed to the

components of wine terroir and, among them, to the vineyard microbiome communities, as possible and previously neglected new key determinants of terroir features that are associated with geographical location and are reported to be directly relevant to vine growing, grape quality and winemaking[6,7]. Indeed, a reliable biological signature of the vineyard microbiome depending on the geographical location of the vineyard has recently been demonstrated[8], but little is known about its variations at finer local spatial scales[9], possibly matching different PDO areas, particularly in terms of the local diversity of plant growth-promoting (PGP) micro-organisms as determinants of growth promotion, yield enhancement, and product quality[7]. In this context, we hypothesized that the interplay between bulk and rhizospheric soil microbiomes may represent an integral component of terroir, influencing nutrient uptake, and the overall terroir expression in defining the unique qualities of vineyards. Thus, the fine characterization of bulk and rhizospheric soil in the different PDO terroirs

[1]Unit of Microbiome Science and Biotechnology, Department of Pharmacy and Biotechnology, University of Bologna, via Belmeloro 6, 40126 Bologna, Italy. [2]Fano Marine Center, The Inter-Institute Center for Research on Marine Biodiversity, Resources and Biotechnologies, viale Adriatico 1/N, 61032 Fano, Pesaro Urbino, Italy. [3]Unit of Enzymology, Department of Pharmacy and Biotechnology, University of Bologna, viale Risorgimento 4, 40136 Bologna, Italy. [4]Laboratorio di Proteomica Metabolomica e Chimica Bioanalitica, IRCCS Istituto delle Scienze Neurologiche di Bologna, Bologna, Italy. [5]Department of Chemistry "G. Ciamician", University of Bologna, Via Selmi 2, 40126 Bologna, Italy. ✉e-mail: simone.rampelli@unibo.it

may provide important highlights on the relevance of local soil microbiome diversity in defining the distinct organoleptic characteristics of wines from specific regions[10].

To provide some insights in this direction, here we aimed to investigate possible differences in microbiome-dependent terroir characteristics (rhizospheric and bulk soil microbiomes) in plant samples of *Vitis vinifera* cultivar Sangiovese collected from 12 different sub-areas located within the "Consorzio del Vino Nobile di Montepulciano DOCG" PDO area, in Tuscany, Italy. In particular, Montepulciano and its territory are considered an excellence in the Italian food and wine context, with the "Vino Nobile di Montepulciano" renowned all over the world, with 7 million bottles sold and a production turnover of 65 million euros in 2022, for a total estimated value of around 1 billion euros, including the value of assets (https://www.ansa.it/canale_terraegusto/notizie/vino/2023/02/15/vino-nobile-montepulciano-distretto-vale-1-mld-di-euro_14425b81-3f63-4d41-b29a-db1469fbed30.html). Montepulciano territory has recently been divided into 12 production areas (i.e., additional geographical units—AGUs), called "Pievi", each of them showing different characteristics in terms of altitude, pedoclimatic characteristics, soil composition and chemistry (https://www.doctorwine.it/en/pot-pourri/miscellanea/the-nobile-revolution-pieve, last access February 2024). The possibility of subdividing the production area was also made possible by the fact that the wines exhibited different organoleptic profiles, which reflected the specific characteristics of the terroir. This paved the way for the characterization of the microbiome determinants of this territorial uniqueness. In particular, we proposed a finer characterization of the microbial terroir within the 12 AGUs in order to add a microbiome dimension to the terroir features, to better understand and thus safeguard the local diversity of Italian wine production. In addition to enriching our understanding of the importance of soil and root-associated microbiomes in defining wine terroir within the Vino Nobile di Montepulciano PDO area, this study may provide further economic incentives for agricultural and enological practices that preserve regional microbial terroir and biodiversity.

## Results

### Microbial characteristics of viticultural terroirs of *V. vinifera* cultivar Sangiovese for the production of Vino Nobile di Montepulciano

A total of 336 root samples (rhizosphere) of *V. vinifera* cultivar Sangiovese and 56 bulk soil samples were collected from 14 different vineyards in the 12 AGUs in July, August, September, and October 2022 in Montepulciano (Tuscany), Italy (Fig. 1). Specifically, for each vineyard, 6 rhizospheric samples and 1 bulk soil sample were retrieved at each time point. All selected vineyards were located within the PDO area. Information on sites and plant characteristics, rootstock families, and management, as well as physical and chemical variables of vineyards soils, are provided for each AGU in Supplementary Tables 1 and 2, respectively. For the 12 AGUs and the four-time points, the composition of the soil and rhizosphere microbiomes was first investigated by next-generation sequencing of the bacterial 16S rRNA gene (V3–V4 hypervariable regions) and fungal ITS (internal transcribed spacer ITS2 region), with 332 (292 rhizosphere and 40 soil) and 64 (50 rhizosphere and 14 soil) samples successfully sequenced, respectively. This resulted in 3,654,656 high-quality reads, with an average of $11,008 \pm 4723$ reads per sample (mean ± SD), for 16S rRNA gene sequencing data, and in 382,144 high-quality reads ($5971 \pm 3240$) for ITS sequencing data. Reads were binned into 57,395 amplicon sequence variants (ASVs) for 16S rRNA gene sequencing and 740 ASVs for ITS sequencing.

In order to identify the soil microbiome peculiarities of the microbial terroir within the "Consorzio del Vino Nobile di Montepulciano DOCG", we compared its bacterial and fungal composition with bulk soils from vineyards from all over the world, including Chile, Argentina, USA, South Africa, Australia, Spain, France, Italy, Hungary, Portugal, Denmark, Germany and Croatia[8] (Fig. 2). We observed the effect of geographical distance on the composition and structure of soil microbial communities, both bacterial and fungal, with individual countries significantly segregating in the principal coordinates analysis (PCoA) plots (permutation test

**Fig. 1 | Map of the 12 production areas (i.e., additional geographical units—AGUs) recognized by the "Consorzio del Vino Nobile di Montepulciano DOCG" (Tuscany, Italy).** The production areas are indicated by different colors with the names in bold, Valiano, Valardegna, San Biagio, Sant'Albino, Le Grazie, Gracciano, Cervognano, Cerliana, Caggiole, Badia, Ascianello and Sant'Ilario (map source Consorzio del Vino Nobile di Montepulciano).

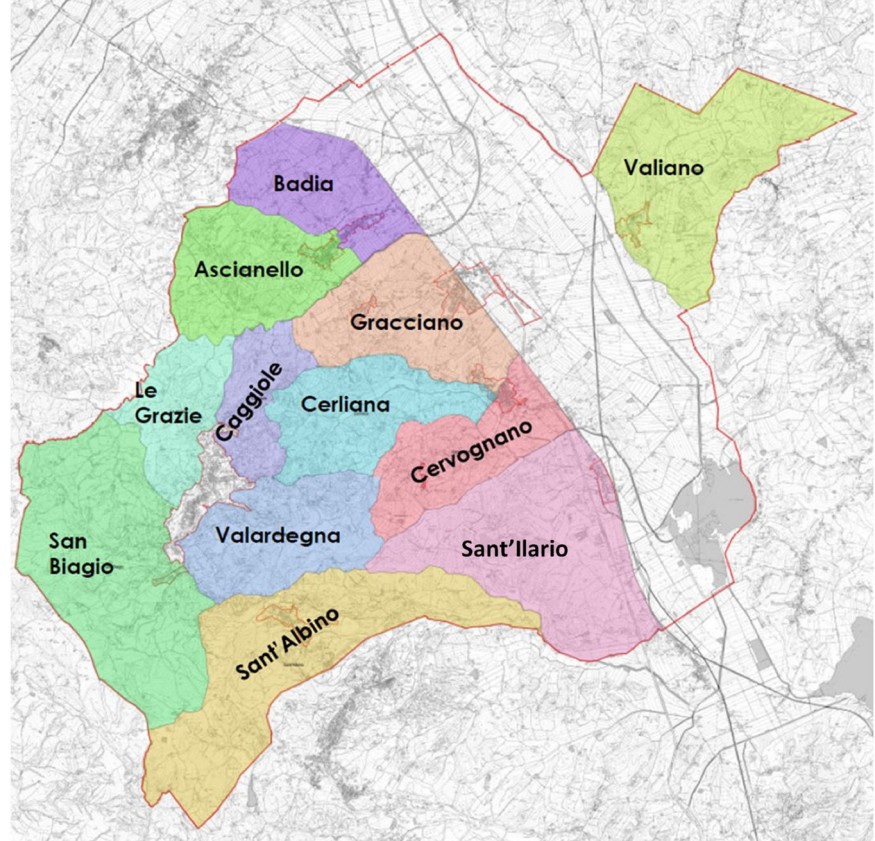

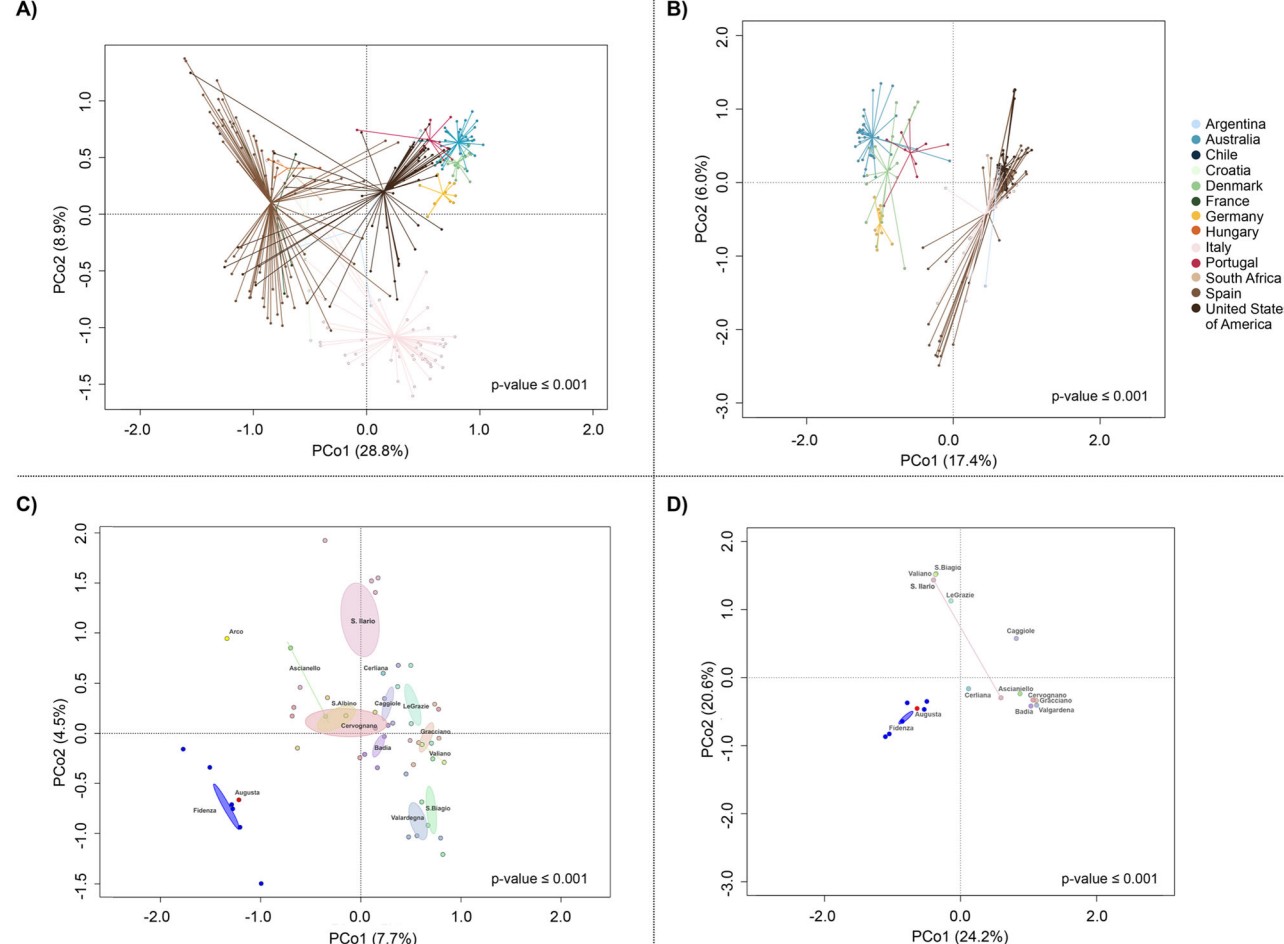

**Fig. 2 | The bulk soil microbiome of Montepulciano vineyards shows a clear differentiation compared to other vineyards around the world.** Comparisons were made for both 16S rRNA and ITS sequencing, using data from Gobbi et al.[8]. **A** Principal coordinate analysis (PCoA) based on unweighted UniFrac distances showing the variation of *Vitis vinifera* cultivar Sangiovese bulk soil bacterial composition at a wide geographical scale (worldwide), including Montepulciano samples in the Italian site (permutation test with pseudo-*F* ratio, *p*-value ≤ 0.001). **B** PCoA based on Bray–Curtis distances showing the variation of *V. vinifera* cultivar Sangiovese bulk soil fungal composition at a wide geographical scale (worldwide), including Montepulciano samples in the Italian site (*p*-value ≤ 0.001). **C** The same graph as in (**A**), but at a finer geographical scale, including only Italian samples from Gobbi et al.[8] and Montepulciano samples (*p*-value ≤ 0.001). **D** The same graph as in (**B**), but at a finer geographical scale, including only Italian samples from Gobbi et al.[8] and Montepulciano samples (*p*-value ≤ 0.001). For **C** and **D**, sample origin is indicated on each graph. For all PCoA plots, the first and second principal components are plotted and the percentage of variance in the dataset explained by each axis is shown.

with pseudo-*F* ratio, *p*-value ≤ 0.001) (Fig. 2A and B). At the national scale, i.e., considering only bulk soil samples from Montepulciano and other Italian vineyards, we also observed a significant segregation of vineyards according to region of origin (*p*-value ≤ 0.001) (Fig. 2C and D).

When investigating the soil microbial taxa responsible for the geographical segregation, we identified five bacterial genera (Fig. 3A) and 5 fungal genera (Fig. 3B) whose variation in relative abundance was significantly different between Montepulciano and any other vineyard in Italy and worldwide (Kruskal–Wallis test controlled for multiple testing using false discovery rate—FDR, *p*-value ≤ 0.05; refer to Supplementary Data 1 for original data for Fig. 3 production). Specifically, for bacterial taxa, we found that the genera *Ilumatobacter*, *Microlunatus*, and *Hydrogenispora* were almost exclusively present in the Montepulciano consortium, while *Gemmata* and *Nocardioides*, widely distributed in the different soils, characterized the Montepulciano area in terms of relative abundance. As for fungal taxa, the genera *Rhizopus*, *Gongronella*, *Lipomyces*, and *Penicillium* were almost exclusively present in the Montepulciano consortium, while *Mortierella* characterized the Montepulciano soil in terms of relative abundance.

We then tried to define a core soil microbiome of the Vino Nobile di Montepulciano area, looking for taxa present in the bulk soil of all AGUs.

We identified five microbial genera with this characteristic, namely *Nocardioides*, *Solirubrobacter*, *Gemmatimonas*, *Haliangium*, and *Pirellula*. Interestingly, *Nocardioides* was both a core taxon and a genus that distinguished the Montepulciano territory from vineyards in the rest of the world, and for this reason, it could be considered the main marker characterizing the microbial terroir of Vino Nobile. Interestingly, these core genera were also present in all 12 AGUs when considering the rhizospheric soil, indicating a continuity between soil and rhizosphere in the Montepulciano territory. This continuity was further confirmed with a Procrustes correlation test using the protest function in R, comparing the beta diversity distribution of soil and rhizospheric samples and resulting in a significant correlation (*p*-value = $2*10^{-4}$ for bacterial community and *p*-value = 0.01 for fungal community).

## Spatial distance determines the similarity of microbial communities in vineyards at local scales across the Montepulciano territory

Aware of the continuity between soil and rhizosphere microbiomes, as shown in the previous section, we then aimed to identify the specificities of microbial terroir associated with the recent zonation in the 12 different AGUs of the Montepulciano territory, considering both bacterial and fungal

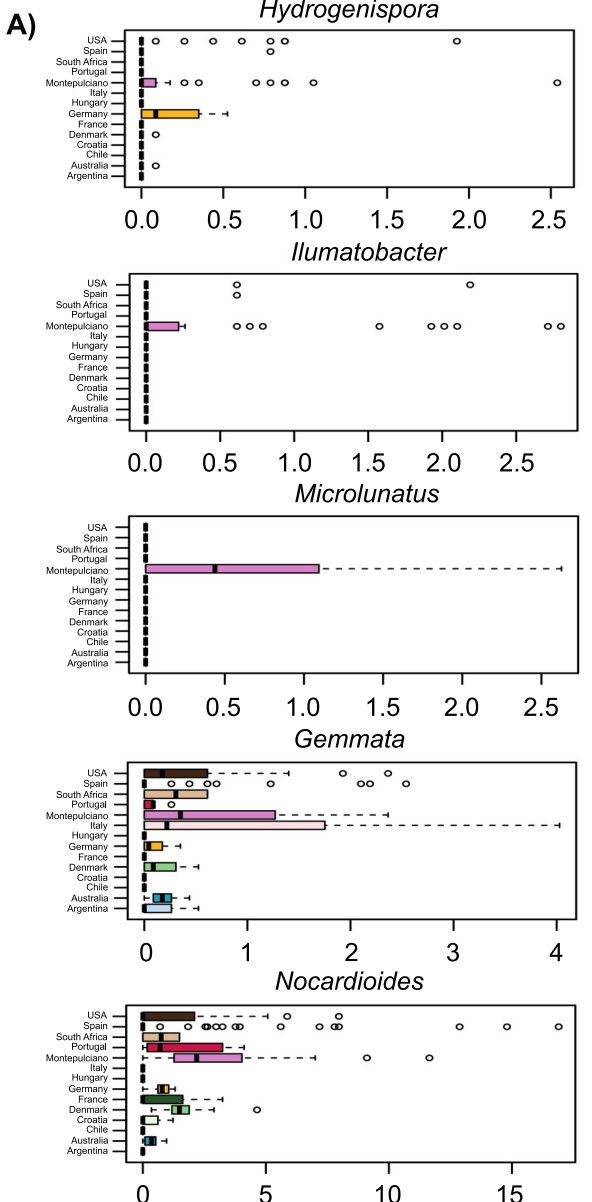

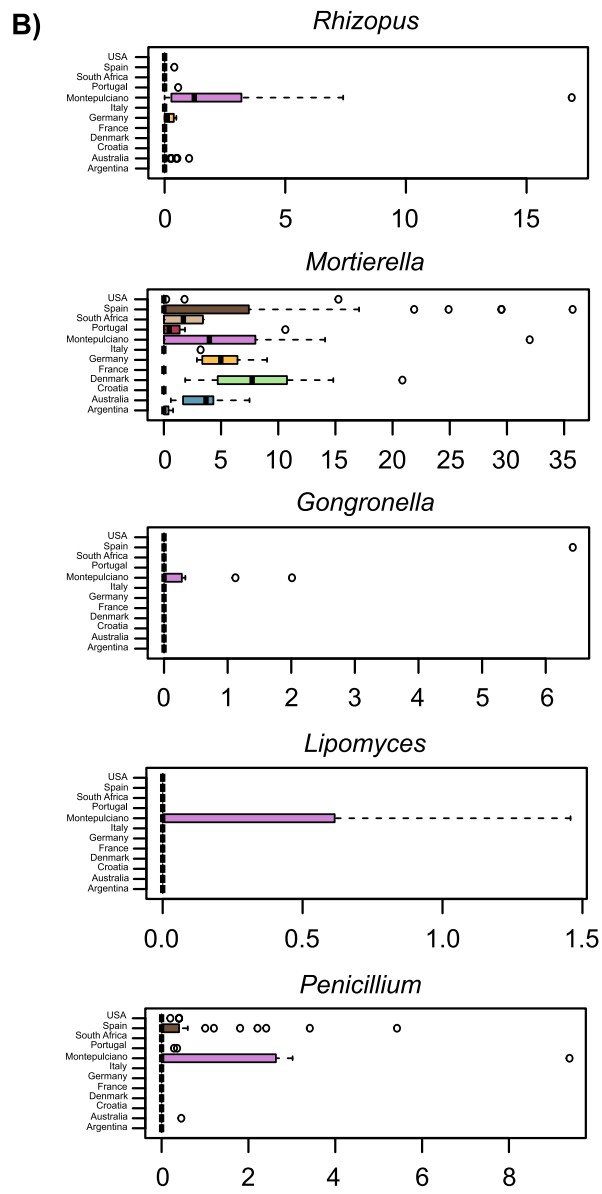

**Fig. 3 | Microbial taxa distinguishing bulk soil samples of the Montepulciano territory from other vineyards worldwide.** Boxplots showing the relative abundance distribution of bacterial (**A**) and fungal (**B**) genera differentially represented in bulk soil between Montepulciano and other vineyards worldwide (from Gobbi et al.[8])

(Kruskal–Wallis test controlled for multiple testing using FDR with $n = 178$ independent samples, $p$-value $\leq 0.05$; refer to Supplementary Data 1 for original data for Fig. 3 production).

counterparts. We found that the differences in the rhizosphere microbiome were explained by the geographical distance between the different AGUs, with the AGUs of Sant'Ilario (southeast of the territory) and San Biagio (west), located on opposite borders of the territory, having the most different bacterial and fungal configurations, and other AGUs having intermediate configurations between the two extremes. This segregation pattern was robust to seasonality, agronomical practices and management, vine clone type, rootstock family, altitude, and soil composition (permutation test with pseudo-$F$ ratio, $p$-value $\leq 0.01$) (Fig. 4). Specifically when comparing microbiomes across time points, we found the same segregation as if the main factor driving microbiome differentiation was geographical origin at a very local scale (AGUs) rather than plant maturity and season (Procrustes test, $p$-value $\leq 0.01$) (Supplementary Fig. 1). In support of this evidence, we also found that rhizosphere microbiome separation in the PCoA correlated with geographical separation in terms of distance (in meters) between vineyards ($p$-value $\leq 0.003$). AGUs also showed a different alpha-diversity

configuration among them (Supplementary Fig. 2). However, we did not observe a common pattern based on geographical distribution.

Random forest[11] was then used to identify rhizospheric bacterial and fungal genera that distinguished the 12 AGUs and then combined with the Kruskal–Wallis test among relative taxon abundances in each AGU, to extract as much information as possible from our analysis. All significantly discriminating genera identified were represented as a heatmap using their relative abundance in each AGU (Fig. 5). For the bacterial component of the rhizospheric soil, 24 genera were found to be discriminant among AGUs, 11 of which belonging to uncultured or unassigned genera. As for the fungal counterpart of the rhizospheric soil, six genera were identified as discriminating among AGUs. Such patterns reflected a sort of gradient describing the variation in relative abundance of these microorganisms along the Montepulciano territory, from Sant'Ilario to San Biagio and vice versa, crossing all other AGUs in an intermediate configuration between the two extremes. This was very clear when we superimposed the gradient of

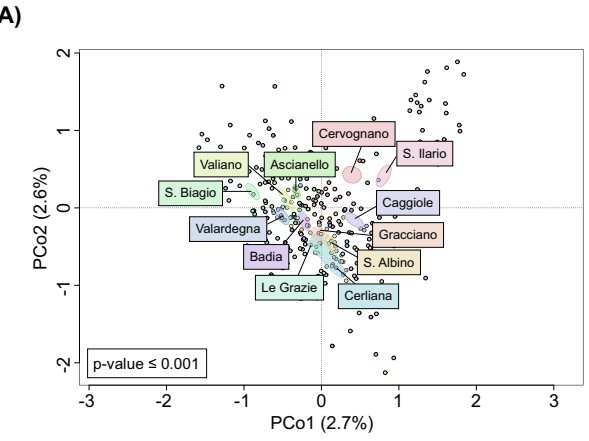

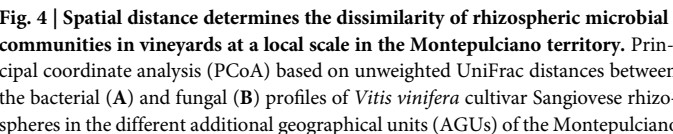

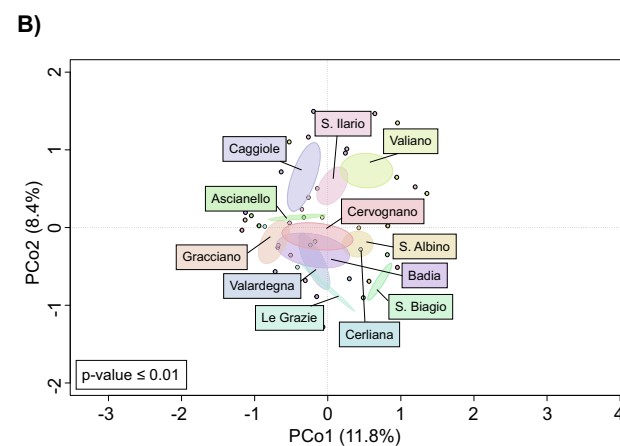

**Fig. 4 | Spatial distance determines the dissimilarity of rhizospheric microbial communities in vineyards at a local scale in the Montepulciano territory.** Principal coordinate analysis (PCoA) based on unweighted UniFrac distances between the bacterial (**A**) and fungal (**B**) profiles of *Vitis vinifera* cultivar Sangiovese rhizospheres in the different additional geographical units (AGUs) of the Montepulciano territory. The first and second principal components are plotted, and the percentage of variance in the dataset explained by each axis is shown. *P*-values are calculated with a permutation test with a pseudo-*F* ratio, taking into account the contribution of seasonality, agronomical practices and management, vine clone type, rootstock family, altitude, and soil composition (*p*-value ≤ 0.01).

relative abundance of microorganisms on the map of the territory (Fig. 6). Interestingly, these characteristics of the rhizosphere were confirmed at the level of soil microbiome (Supplementary Figs. 3 and 4).

**Understanding the functional peculiarities of the microbial terroir in the Vino Nobile di Montepulciano PDO area**

We performed shotgun metagenomics on a subset of 28 samples, one bulk soil sample and one rhizosphere sample for each vineyard, representative of each Montepulciano AGU at the first time point, to obtain a more accurate picture of the pattern of variation of bacterial PGP functions across the Vino Nobile di Montepulciano PDO area. We retained ~390$M$ high-quality reads, with an average of 14$M$ ± 9$M$ (mean ± SD) paired-end sequences per sample.

Reads were first aligned to known PGP genes[9] to screen for the potential microbial ability to support plant growth at the soil-root interface. The microbial PGP functions selected for our analysis were: (i) nitrogen (N) fixation; (ii) phosphorous (P) solubilization; (iii) iron chelation; (iv) production of the phytohormone indole-3-acetic acid (IAA); and (v) production of the enzyme 1-aminocyclopropane-1-carboxylic acid (ACC) deaminase. We found that the rhizosphere microbiome of each AGU showed its own peculiar functional profile of microbial PGP traits, with the AGUs of the southeastern area showing an overall greater potential for P solubilization, while those of the western area showing a greater propensity for ACC deaminase production (*p*-value = 0.05, Wilcoxon test, Fig. 7). It is also interesting to note that the functional potential for siderophore and IAA production showed an isolated peak in the northern and central part of the territory, respectively. Finally, the N fixation potential was more homogeneous in the production area (z-score between −1 and 1), although it was more present in the AGUs of Sant'Ilario, Gracciano, Cerliana, and Ascianello. Notably, all PGP functions were present at similar levels in the soil microbiomes of the corresponding AGUs, again supporting the continuity between the two ecosystems also from a functional point of view (r > 0.98, *p*-value < 0.0001, Pearson's correlation) (Supplementary Table 3).

We then used the entire set of 28 metagenomes to reconstruct metagenome-assembled genomes (MAGs), with the aim of increasing the taxonomic resolution of 16S rRNA gene sequencing analysis and matching the functional potential to the corresponding taxonomy. We were able to reconstruct 37 MAGs with more than 50% completeness and <5% contamination, 15 of which were taxonomically assigned to the previous AGU-associated bacterial taxa and 4 to the core bacterial genera of the production area (Supplementary Table 4). Specifically, of the 15 MAGs assigned to AGU-associated bacterial genera, three were assigned to unclassified species

of Conexibacteriaceae, one to *Massilia yuzhufengensis*, one to *Bradyrhizobium algeriense*, one to unclassified species of the genus *Mycobacterium*, one to *Nocardioides iriomotensis*, two to unclassified species of the genus *Nocardioides*, one to unclassified species of the genus *Sphingomonas*, two to *Steroidobacter denitrificans*, and three to unclassified species of the genus *Streptomyces*. The four MAGS assigned to core genera included the three MAGs assigned to *Nocariodes* and one to unclassified species of the genus *Solirubrobacter*.

The entire set of MAGs was further processed by directly aligning them to the previous PGP gene sequences used to screen for the potential ability to support plant growth at the soil–root interface in each AGU. This analysis was integrated by applying METABOLIC[12], a software that computes the contribution of microorganisms to biogeochemical transformations and cycles of carbon, N, and sulfur (Fig. 8 and Supplementary Table 5). Looking specifically at the MAGs assigned to the core taxa, we found that they covered a very broad range of functions capable of supporting soil fertility and plant health. In particular, the two MAGs assigned to unclassified *Nocardioides* (bin.197 and bin.92) and the one assigned to *N. iriomotensis* (bin.111) encoded for genes involved in siderophore production, N fixation, nitrite ammonification, nitrate reduction, iron reduction, organic substrate fermentation, acetate oxidation, and organic carbon oxidation. In addition, the MAG assigned to *Solirubrobacter* (bin.178) carried the genes necessary for the oxidation of organic carbon from amino acids and complex carbohydrates, including several glycosyl hydrolases, such as GH5, GH65, GH113, GH39, and GH15, which are involved in the degradation of various polysaccharides, such as mannans and glucans.

We then analyzed the MAGs assigned to AGU-discriminant taxa for PGP functions previously shown to be discriminant for the different AGUs, specifically P solubilization, which was higher in AGUs on the southeastern side of the territory, and ACC deaminase production, which was higher in AGUs on the western side. Among the MAGs characterizing the AGUs in the southeastern side of the territory, we found that bin.348 (*Conexibacter*) was equipped with metabolic pathways for P solubilization, while bin.126 (*Streptomyces*), characterizing the AGUs in the western side, carried the genes necessary for ACC deaminase production. Looking more widely at the functional features characterizing the 15 AGU-related MAGs, we found that the two MAGs for *S. denitrificans* (bin.123 and bin.42), the three for *Conexibacter* (bin.1, bin.176 and bin.348) and the one for *B. algeriense* (bin.340), whose higher abundances were associated with the AGUs in the southeastern part of the territory, encoded the functions necessary for IAA production, sulfur oxidation, N fixation, nitrate reduction, nitrite ammonification, nitrite

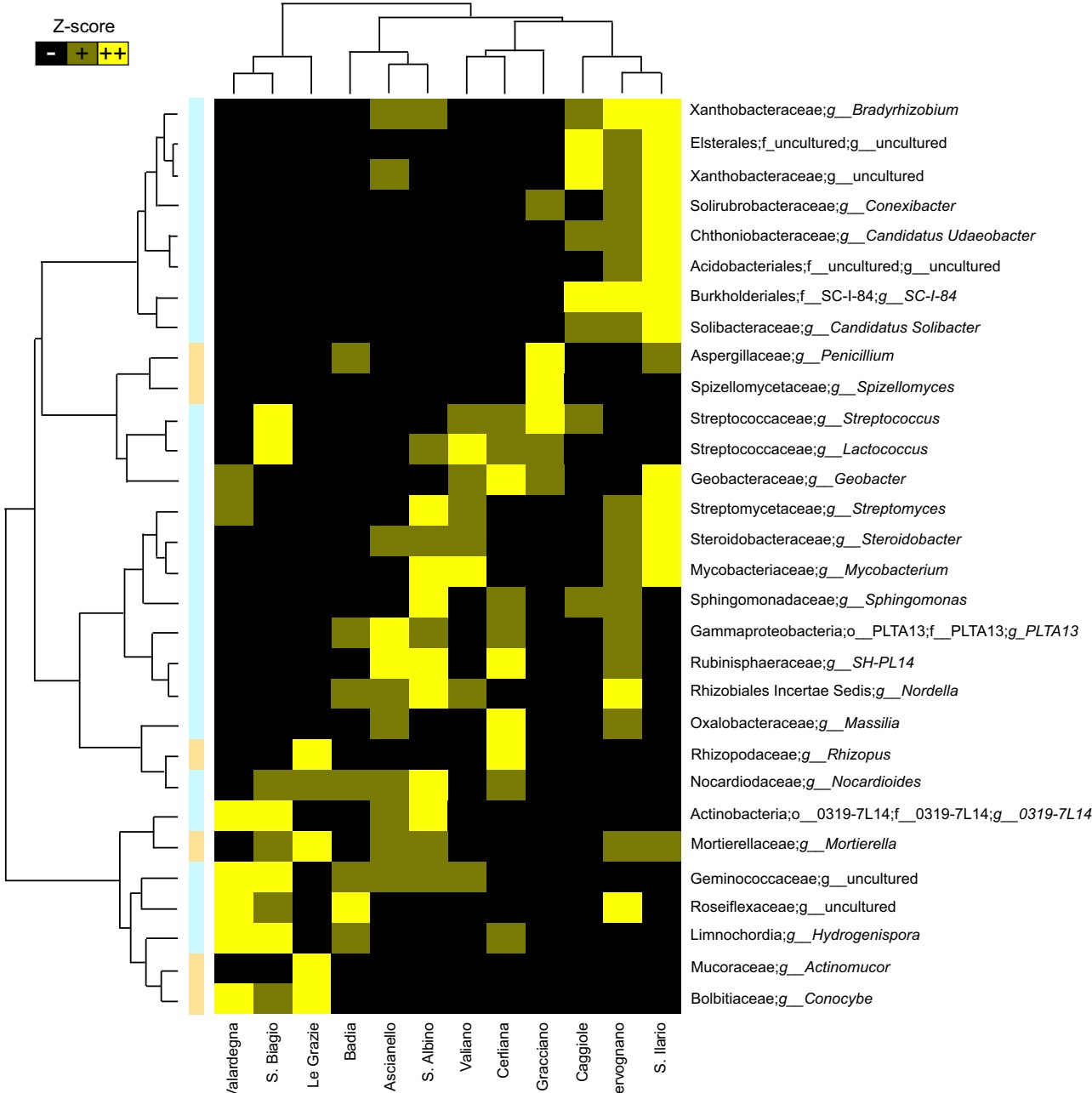

**Fig. 5 | The AGU-related taxa show a relative abundance gradient in the Montepulciano area, describing the rhizospheric microbial variation across the 12 additional geographical units (AGUs).** Heatmap showing all significantly discriminating genera among the rhizosphere microbiomes of the 12 AGUs (random forest combined with the Kruskal–Wallis test among relative taxon abundances in each AGU, $p$-value ≤ 0.05). Relative taxon abundance is represented in the heatmap through $z$-score. The vertical bar is colored blue for bacteria and orange for fungi.

reduction, fermentation of organic substrates, iron oxidation and reduction, siderophore production, thiosulfate oxidation, fermentation of organic compounds, acetate oxidation and oxidation of organic carbon from different sources including amino acids, fatty acids and complex carbohydrates. Furthermore, the MAG assigned to *M. yuzhufengensis* (bin.228), characteristic of the AGUs in the central part of the territory, carried the genes devoted to siderophore production. Finally, the three MAGs assigned to the core taxon *Nocardioides* (bin.111, bin.197 and bin.92) were more abundant in the AGUs in the southern part of the territory, as were the three MAGs assigned to *Streptomyces* (bin.126, bin.173 and bin.43). In particular, the latter encoded the genes necessary for ACC deaminase and siderophore production, N fixation, iron reduction, fermentation of organic molecules, sulfur oxidation and

oxidation of organic carbon from different sources such as complex carbohydrates and aromatic compounds.

## Wine metabolomics highlight association between product features and the variation of the microbial terroir in the different AGUs

In order to explore matches between wine characteristics and the variation of the microbial terroir in the different AGUs, a metabolomic analysis of the wines of 2022 vintage (the year of the sampling campaign) from wineries where grapes were taken exclusively from the same AGU (namely, Sant'Ilario, Caggiole, Cervognano, Le Grazie, Valardegna and Valiano) was conducted. Interestingly, the diversity of the wine metabolic profiles

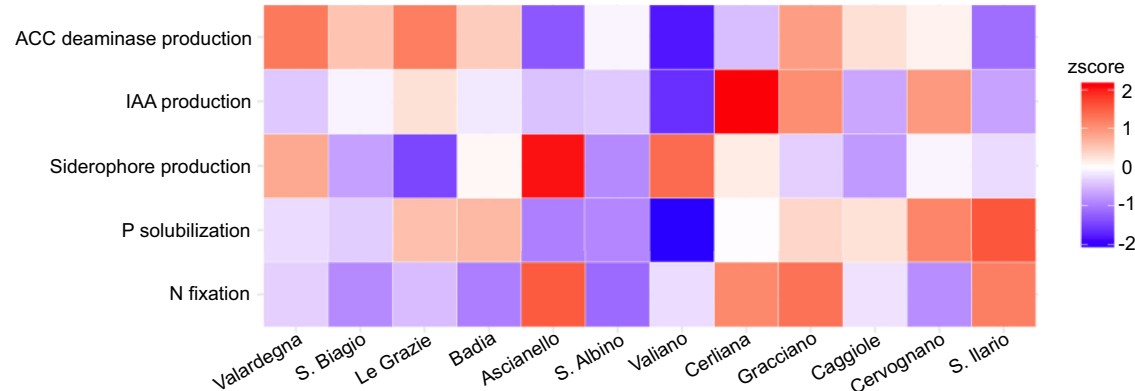

**Fig. 6 | The AGU-related taxa show a pattern of relative abundance variation across the 12 additional geographical units (AGUs).** Maps of the relative abundance of discriminating microbial components of the rhizosphere, both bacterial and fungal. The AGU map is shown at the bottom right for reference to the AGU location, together with a color code for the relative abundance percentage (r.a. %). All maps were created using the QGIS open-source tool (https://www.qgis.org/it/site/).

**Fig. 7 | Functional plant growth-promoting (PGP) profile of additional geographical units (AGU) rhizosphere microbiomes.** Heatmap of PGP functions identified in rhizosphere samples from the different AGUs. PGP functions were normalized in copies per million ((reads a count for an enzyme in a given sample/(gene length/1000))/(no. of reads per sample/$10^6$) and represented in the heatmap through $z$-score. ACC 1-aminocyclopropane-1-carboxylic acid, IAA indole-3-acetic acid, P phosphorous, N nitrogen.

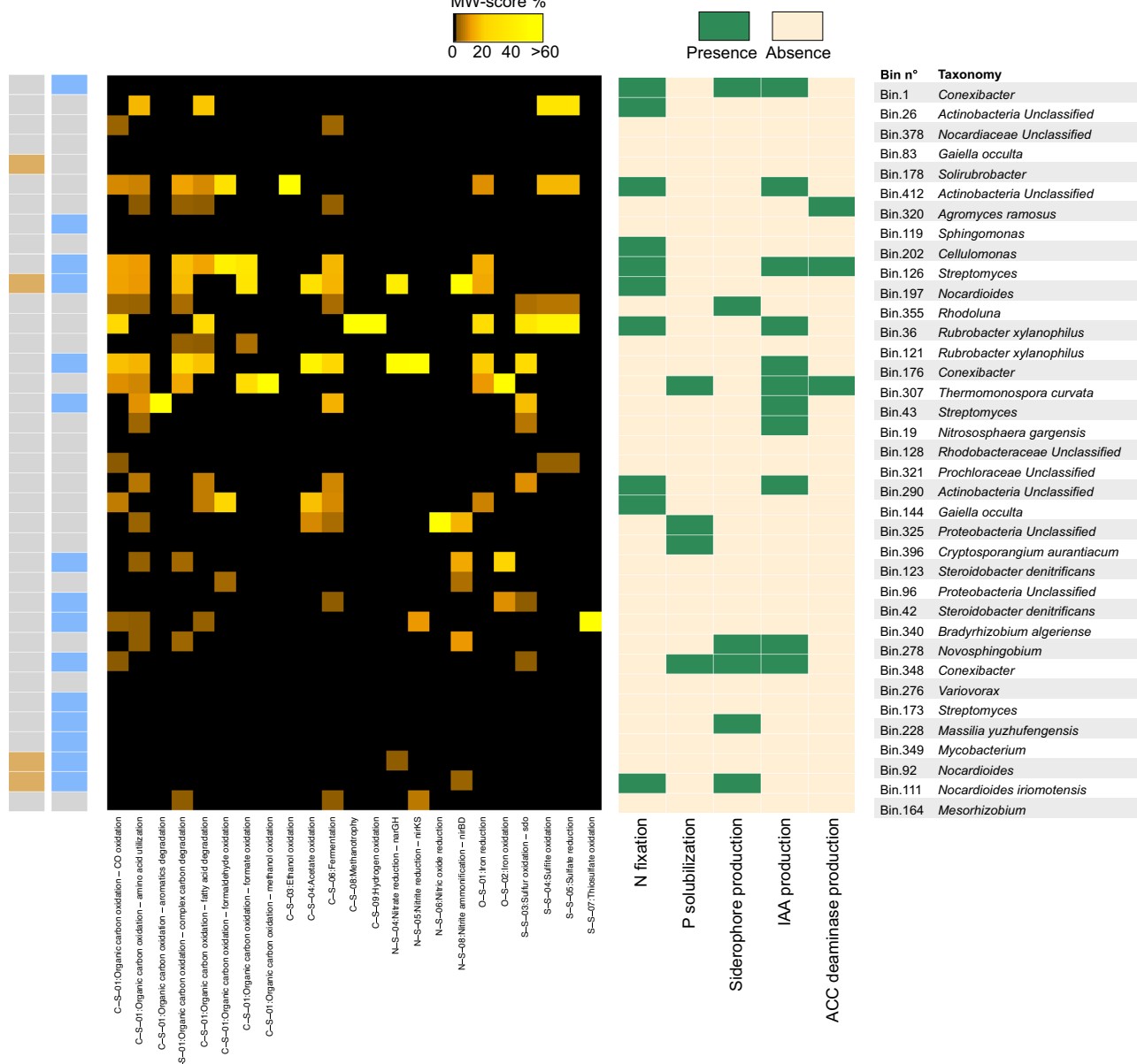

**Fig. 8 | Presence of plant growth-promoting (PGP) functions in reconstructed metagenome-assembled genomes (MAGs) across the Montepulciano territory.** Gradient heatmap of metabolic functions identified in the MAGs and presence/absence table of the PGP functions. From left to right: (i) core taxa identified among MAGs in beige; (ii) MAGs with taxonomy corresponding to discriminant taxa in light blue; (iii) METABOLIC functions identified in the MAGs in a gradient of MW-score percentage (black "0", yellow ">60%"); (iv) PGP pathways identified in the MAGs (green = presence; light beige = absence); and (v) MAGs number and corresponding taxonomy. The MW-score represents the metabolic potential of each MAG within the Montepulciano territory, based on its coverage (how much this MAG is present in the territory), and on the presence or absence of the genes in the MAG (whether the analyzed function is present or not).

matched the variation in the overall rhizosphere microbiome configuration in the corresponding AGU ($p$-value = 0.04, Procrustes test, Supplementary Fig. 5C). In order to identify the wine metabolites responding to changes in the rhizosphere microbiomes in the different AGUs, the analytical components were superimposed on the PCoA plot of the Unweighted UniFrac distances between bacterial and fungal profiles of the rhizospheres microbiomes at the different AGUs. Interestingly, several wine metabolites, such as L-acetylcarnitine, L-methionine, quercetin, and citicoline for the bacterial configuration, and adenine for the fungal configuration, were significantly associated with terroir specificities of the rhizosphere microbiome in the different AGUs ($p$-value < 0.05, "envfit" function in the vegan R package, Supplementary Fig. 5A and B).

## Discussion

In the present study, we characterized the soil–plant microbiome dimension of the variation of the viticultural terroir of *V. vinifera* cultivar Sangiovese from the "Consorzio del Vino Nobile di Montepulciano DOCG" PDO area in Tuscany, Italy. This was made possible by comparing the microbiomes associated with the bulk soil and rhizosphere of vineyards located in the 12 AGUs recently established by the consortium. At first, we explored specificities in the bulk soil microbiome from Montepulciano territory with respect to other vineyards in Italy and around the world. Data revealed a clear differentiation of the Montepulciano bulk soil vineyard microbiomes compared to all other vineyards, with 10 microbial genera characterizing the Montepulciano territory. Specifically, 5 bacterial and 5 fungal taxa were

identified, namely the bacteria *Hydrogenispora*, *Ilumatobacter*, *Microlunatus*, *Gemmata* and *Nocardioides*, and the fungi *Rhizopus*, *Mortierella*, *Gongronella*, *Lipomyces* and *Penicillium*. Among these, *Nocardioides* was also part of the core microbiome of Montepulciano vineyards, together with other four taxa, specifically *Solirubrobacter*, *Gemmatimonas*, *Haliangium*, and *Pirellula*, as they were present in all AGUs. It is noteworthy that the genus *Nocardioides* has recently been reported as one of the beneficial microorganisms capable of countering and preventing *Fusarium oxysporum* infection in crops[13], as well as being a potential carrier of other multiple PGP traits[9]. *Gemmatimonas* and *Haliangium* have also previously been associated with plant growth benefits[14,15].

Confirming previous findings, we then highlighted the continuity between the soil and rhizosphere microbiomes in the Montepulciano territory, with evidence that the microbial specificity of the territory (soil) directly reflects the microbiome configuration at the soil-root interface, potentially determining different interactions that differentially affect plant growth and biology[5].

In order to derive microbiome specificities at the AGU level, we focused on the rhizosphere microbiome, and we found that samples clustered according to sampling location, but not to sampling season. In particular, we observed a west–southeast gradient in the relative abundance of some microbial genera (from Sant'Ilario to Argiano AGUs), which correlated with the geographical distances between AGUs and the pedology of the area, suggesting that previous observations of variation in microbial terroir-associated with a national and regional geographical scale[8] are also valid at a local scale. The taxa identified included 24 bacterial and 6 fungal genera, some of which have been previously associated with different PGP functions[13–29].

To deeply explore these observations, we applied shotgun metagenomics to a subset of samples to investigate the presence of PGP functions that could potentially promote plant growth through soil biofertilization and grapevine biostimulation by enhancing nutrient and water uptake and providing higher resistance to environmental stressors, better plant health, and also probably improved wine organoleptic characteristics, thus contributing to regional terroir[5,30–33]. We found that the potential microbial contribution to the biogeochemical cycles of N and carbon, which are critical for soil fertility and plant health, is widely diffused in the analyzed genomes, with the four MAGs assigned to the core taxa (i.e., *Nocardioides* and *Solirubrobacter* as genera present in all AGUs; notably, *Nocardioides* was both a core taxon and a genus that distinguished the Montepulciano territory from vineyards in the rest of the world), carrying the necessary genes for nitrification and denitrification pathways, as well as organic carbon oxidation and fermentation using different substrates, including complex carbohydrates, such as mannans and glucans, known components of plant cell walls[34,35]. The AGUs of the southeastern part of the production area (the side delimited by Sant'Ilario) showed a greater potential for P solubilization, while those of the western part (delimited by S.Biagio) showed a greater propensity for ACC deaminase production. Consolidating this evidence, we also found that one MAG associated with the southeastern part of the territory, assigned to *Conexibacter*, encoded genes devoted to P solubilization, while MAG associated with the western and southern parts, assigned to *Streptomyces*, carried the ACC deaminase gene. These differences match local peculiarities of the terroir, with the southeastern AGU-associated microbiome providing an extra means of P provision from local soil, which resulted in general depletion of this important nutrient[36], and the western AGU-associated microbiome potentially helping plants to respond to salt and drought stress[37,38]. This set of metabolic potentials, either common to all AGUs or specific to some of them, represents a promising avenue for leveraging microbial terroir as a mediator between soil resources and plant requirements[7], with possible implications on local product quality and productivity, possibly contributing to wine differentiation depending on the AGU of origin. Indeed, by controlling for plant P availability, the root-associated microbiome can influence several sensory characteristics, including the aroma, appearance, flavor, and taste, of its associated wines[39,40]. On the other hand, by counteracting excessive drought stress, root

microbiome ACC deaminases protect against delays in fruit ripening and the consequent loss of varietal character, which is crucial for flavor development[41,42].

To provide insights into possible connections between the observed compositional and functional variations in the microbial terroir at the AGU level and the corresponding organoleptic characteristics of the produced wines, metabolomic profiles of the wines were analyzed. Interestingly, several associations between wine metabolites and the terroir microbiomes were observed across the different AGUs. Specifically, varying concentrations of L-acetylcarnitine, L-methionine, quercetin, citicoline, and adenine in the wines from Sant'Ilario, Caggiole, Cervognano, Le Grazie, Valardegna, and Valiano, respectively, were linked to corresponding AGU-level specificities in the rhizosphere microbiome structure.

These molecules have been previously reported as key determinants of vine characteristics. For instance, L-acetylcarnitine in grapes can influence the synthesis of esters that enhance the wine's aroma profile[43], while L-methionine has been associated with the production of volatile compounds that contribute to the aromatic complexity of wine[44]. Finally, quercetin contributes to the color, flavour, and health benefits of wine[45]. On the other hand, there is no documented evidence in the literature that the presence of different concentrations of citicolines and adenines in wine influences its organoleptic profile. Taken together, these findings suggest a possible connection between local features in the terroir microbiomes in the different AGUs and correspondent organoleptic features of the produced wines.

Our findings, showing the potential relevance of the local diversity of terroir microbiome in maintaining plant health and productivity, and potentially wine product quality, became relevant when placed in the current context of global change, leading to nutrient and soil depletion and loss of microbial diversity[46–49]. In this scenario, the characterization of the root-associated microbiome encoding PGP functions could represent the first step towards new strategies to improve the sustainability and resilience of viticulture, integrating management strategies for the protection and preservation of the local microbial terroir features as a key aspect in the product quality[50].

Our results, coupled with the growing evidence of the significant influence of both soil and plant microbiomes on the sensory properties of the final product[51,52], may lay the foundations for a new perspective in which the local variation of microbiome features in terroir needs to be protected as a biodiversity treasure highly linked to the local diversity of wine production and traditions. This is particularly relevant in cases where the product is closely linked to its geographical origin, such as within a PDO area, like DOCG in Italy, or when the vineyard location is indicated by the term "cru", which immediately links the product to a precise growing location. The local microorganisms carrying the genes for PGP functions could be those best suited and preserved to thrive in the local pedoclimatic conditions and contribute to healthy plant development. This will require the integration of current viticultural strategies with a precise and tailored microbiological approach. It will entail the combination of the isolation of PGP microorganisms with the metagenomic approach, thereby enabling a comprehensive investigation of their functions through genome sequencing and targeted functional assessments. This represents an unexplored frontier aimed at safeguarding and enhancing the properties and qualities of wine in a context of global change by exploiting the natural microbiomes of the vineyard.

## Methods
### Study sites, sample and metadata collection, and sample pre-treatment
Grapevine roots and soil samples were collected from the 12 production areas (AGUs) within the "Consorzio del Vino Nobile di Montepulciano DOCG" PDO area in Tuscany, Italy. All plants were *V. vinifera* cultivar Sangiovese, apparently healthy, more than 15 years old, and used for the production of Vino Nobile di Montepulciano. Sampling was carried out at four different time points throughout the production season in 2022 (i.e., pre-harvest in July, August, and September, and post-harvest in October)

for a total of 336 root samples and 56 soil samples. Specifically, for each of the 14 vineyards, 6 rhizospheric samples and 1 bulk soil sample were retrieved at each timepoint. Chemical features of the vineyard soil (e.g. P and N concentrations) were measured at the time of sample collection using a multiparametric probe. In particular, for the microbial characterization of the rhizospheric soil, thin lateral roots of the grapevine were collected after digging 10–20 cm below the ground surface, whereas bulk soil was collected in the plant proximity at the same depth after removing the surface soil and grass cover if present[9]. All samples were collected using sterile gloves, placed in a sterile 50-ml Falcon tube, transported to the laboratory on ice, and stored at −80 °C until further processing. To separate the rhizospheric soil from the root surface, roots were treated as previously described[9,53]. Briefly, ~3 cm of terminal root portions, including tips, were dissected using sterilized scissors and tweezers. The root segments were then placed in 15-ml Falcon tubes filled with 2.5 ml of modified PBS buffer (130 mM NaCl, 7 mM Na$_2$HPO$_4$, 3 mM NaH$_2$PO$_4$, pH 7.0, and 0.02% Silwet L-77) and left on a shaking platform at 180 rpm for 20 min for washing. Roots were removed from the tubes and the washing buffer was centrifuged at $1500 \times g$ for 20 min, with the resulting pellet regarded as the rhizospheric soil. Metadata, such as agronomical practices and management, vine clone type, rootstock family, and altitude, were collected during the sampling campaigns by inspecting the vineyards and asking the winemakers directly. Soil composition was retrieved from a previous publication[54].

## Microbial DNA extraction and sequencing

Prior to DNA extraction, rhizosphere and soil samples for fungal analysis were treated with 500 μl of lyticase solution (for 1 ml of solution: 978 μl Tris–EDTA, 2 μl β-mercaptoethanol, and 20 μl lyticase 10 U/ml) and incubated at 37 °C for 30 min to facilitate fungal cell wall lysis and nucleic acid recovery. Microbial DNA was extracted from the rhizospheric soil after the pre-treatment described above and from 0.25 g of bulk soil using the DNeasy PowerSoil Pro Kit (QIAGEN, Hilden, Germany) following the manufacturer's instructions with a few modifications. Briefly, the homogenization step was performed using a FastPrep instrument (MP Biomedicals, Irvine, CA, USA) with a cycle consisting of three 1-min steps at 5.5 movements per sec with 5-min incubation on ice between each step. At the end of the protocol, DNA elution was preceded by a 5-min incubation on ice. The resulting DNA was quantified using a NanoDrop ND1000 spectrophotometer (NanoDrop Technologies, Wilmington, DE, USA) and diluted in PCR-grade water to a final concentration of 5 ng/μl before amplification.

For characterization of the bacterial fraction of the rhizospheric and bulk soil microbiome, the V3–V4 hypervariable regions of the 16S rRNA gene were PCR-amplified in a final volume of 50 μl containing 25 ng of genomic DNA, 2X KAPA HiFi HotStart ReadyMix (Roche, Basel, Switzerland) and 200 nmol/l of 341 F (S-D-Bact-0341-b-S-17, 5′-CCTACGGGNGGCWG CAG-3′) and 785R (S-D-Bact-0785-a-A-21, 5′-GACTACHVGGGTATC TAATCC-3′) primers[55] carrying Illumina overhang adapter sequences. The thermal cycle consisted of an initial denaturation at 95 °C for 3 min, followed by 25 cycles at 95 °C for 30 s, 55 °C for 30 s, and 72 °C for 30 s, with a final extension step at 72 °C for 5 min[56]. For characterization of the fungal component, ITS2 was PCR-amplified as above using ITS3 and ITS4 primers[57] carrying Illumina overhang adapter sequences. The thermal cycle consisted of an initial denaturation at 95 °C for 3 min, followed by 30 cycles at 95 °C for 30 s, 56 °C for 30 s and 72 °C for 1 min, with a final extension step at 72 °C for 5 min. All PCR amplicons were cleaned up with Agencourt AMPure XP magnetic beads (Beckman Coulter, Brea, CA, USA) and indexed libraries were prepared by limited-cycle PCR using Nextera technology. Indexing was followed by a second clean-up as described above. Libraries were then quantified using a Qubit 3.0 fluorimeter (Invitrogen, Waltham, MA, USA), normalized to 4 nM, and pooled. Prior to sequencing, the sample pool was denatured with 0.2 N NaOH and diluted to 4.5 pM with a 20% PhiX control. Sequencing was performed on an Illumina MiSeq platform using a 2 × 250-bp paired-end protocol, according to the manufacturer's instructions (Illumina, San Diego, CA, USA).

A subset of 28 representative samples (14 bulk soils and 14 rhizospheric soils) were further processed for shotgun sequencing. DNA libraries were prepared using the QIAseq FX DNA Library Kit (QIAGEN) according to the manufacturer's instructions. Shortly, 100 ng of each DNA sample was fragmented to 450-bp size, end-repaired, and A-tailed fragments using FX enzyme mix with the following thermal cycle: 4 °C for 1 min, 32 °C for 8 min, and 65 °C for 30 min. Adapter ligation was performed by incubating DNA samples at 20 °C for 15 min in the presence of DNA ligase and Illumina adapter barcodes. A first purification step with Agencourt AMPure XP magnetic beads (Beckman Coulter) was performed, followed by library amplification with limited-cycle PCR according to the manufacturer's instructions, and a further purification step. Samples were pooled at an equimolar concentration of 4 nM to obtain the final library. Sequencing was performed on an Illumina NextSeq platform using a 2 × 150-bp paired-end protocol, following the manufacturer's instructions (Illumina). All sequencing reads were deposited in the ENA archive under the accession number PRJEB75007.

## Bioinformatic analysis of microbiome data

For marker gene analysis (16S rRNA gene and ITS), samples were processed using a pipeline combining PANDAseq[58] and QIIME 2[59]. The "fastq filter" function of the Usearch11 algorithm[60] was applied to retain high-quality reads (min/max length = 350/550 bp). Based on the phred Q score probabilities, reads with an expected error per base $E = 0.03$ (i.e., three expected errors every 100 bases) were discarded. The retained reads were then binned into ASVs using DADA2[61]. Taxonomic assignment was performed using the VSEARCH algorithm[62] and the SILVA database[63] (December 2017 release) for bacteria and the UNITE database[64] (May 2021 release) for fungi. For the bacterial analysis, all sequences assigned to eukaryotes (including mitochondrial and chloroplast sequences) or unassigned were discarded, whereas for the fungal analysis, all sequences not assigned to the fungal kingdom were discarded. Normalization was performed to the lowest number of reads for all samples. Beta diversity was estimated by computing unweighted UniFrac distances and Bray–Curtis distances for bacterial and fungal communities, respectively.

In order to identify the soil microbiome peculiarities of the bacterial and fungal composition within the "Consorzio del Vino Nobile di Montepulciano DOCG" with bulk soils from vineyards from all over the world, we retrieved sequencing data from the study of Gobbi et al.[8]. Although Gobbi and co-workers[8] applied a different set of primers for bacterial and fungal analysis, the same bioinformatic pipeline as described above was applied for sequence processing. In order to avoid bias in ASVs assignment, giving the different DNA regions taken into account, we compared Gobbi's and our dataset at genus levels.

All statistical analyses were performed using the R software (R Core Team; www.r-project.org—last accessed March 2021), v. 4.1.2, with the packages "Made4"[65], "vegan" (https://cran.r-project.org/web/packages/vegan/index.html, v2.6-6.1), and "heatmap3"[66]. Data separation in the PCoA was tested using a permutation test with a pseudo-$F$ ratio (function "adonis" in the vegan package). The Wilcoxon rank-sum test and the Kruskal–Wallis test were used to assess significant differences in relative taxon abundance between groups. $P$-values were corrected for multiple testing using the Benjamini–Hochberg method, with an FDR ≤ 0.05 considered statistically significant. Specifically, we used the Procrustes test to compare microbiomes across time points, taking into account seasonality and management practices (i.e., agronomical practices and management, vine clone type, rootstock family, altitude, and soil composition) as well. Random forest[11] with default parameters was used to assess discriminant taxa between AGUs. For the graphical representation of Figs. 4B and 5, we used the soil ITS configuration of Cerliana AGU. In order to confirm soil-rhizospheric continuity of microbial community composition, we performed a Procrustes correlation test on the beta distribution using the "protest" function in R. Variations in wine metabolites related to the microbiome configurations were estimated by correlation analysis calculated using the "envfit" function in the vegan R package.

QGIS software (https://qgis.org/it/site/, last accessed February 2024) was used to construct maps of bacterial and fungal distribution in the Montepulciano territory based on relative taxon abundance. The geographical coordinates of longitude and latitude were used to plot the exact sampling locations in the software. The distribution of relative abundance across samples was obtained using the Triangulated Irregular Network interpolation method in QGIS (TIN interpolation).

For shotgun metagenomics sequencing, KneadData (https://github.com/biobakery/kneaddata, v0.10.0) was used with default parameters to trim and remove low-quality ($q < 20$) reads, tandem-repeated sequences (based on fastqc output) and host reads (*V. vinifera* RefSeq id: GCF_000003745.3). High-quality reads were assembled with MegaHit[67] and the resulting contigs were processed with MetaWRAP[68] for MAG generation. Bins were evaluated for completeness and contamination with CheckM[69]. Only MAGs with >50% completeness and <5% contamination were retained for subsequent analyses. Taxonomic assignment of MAGs was performed with PhyloPhlAn 3.0[70]. All MAGs were tested for their ability to support plant growth by searching for known PGP genes[9]. The microbial PGP functions selected for our analysis were: (i) N fixation (i.e., the genes NifB, NifE, NifH, NifN, NifV, and NifU); (ii) P solubilization (i.e., the alkaline phosphatase phoA and the glucose dehydrogenase GDH); (iii) iron chelation (i.e., the bacterial siderophores-encoding genes EntF/EntS for enterobactin and FslA for rhizoferrin); (iv) production of the phytohormone IAA (i.e., three genes directly involved in IAA synthesis, namely ipdC, aro10, and aldH); and (v) production of the enzyme ACC deaminase (i.e., AcdS gene encoding the enzyme). The amino acid sequence of selected PGP proteins, obtained from the reference sequence of the NCBI protein database (https://www.ncbi.nlm.nih.gov; last access May 2023) (Supplementary Table 6), was recovered and blasted against the MAG sequences using BlastP[71]. Alignments were filtered for query coverage >40% and identity percentage >20%. Reads count for each enzyme within the PGP functions was normalized in copies per million ((reads count for an enzyme in a given sample/(gene length/1000)))/(no. of reads per sample/$10^6$).

Finally, the entire set of MAGs was processed using the METABOLIC software[12] with default parameters to compute their contribution to biogeochemical transformations and cycles of carbon, N, and sulfur. The software calculates the MW-score percentage, which was used for the color gradient of the heatmap in Fig. 8 to show the metabolic potential of each MAG within the Montepulciano territory, based on its coverage (how much this MAG is present in the territory) and the presence or absence of the genes in the MAG (whether the analyzed function is present or not).

## Wine analytical characterization

All wine samples underwent two types of analytical characterization: untargeted headspace solid-phase microextraction gas chromatography-electron impact mass spectrometry (HS-SPME-GC-Orbitrap) and untargeted metabolomics analysis using micro liquid chromatography–high-resolution mass spectrometry (microLC–ESI-QTOF) in data-dependent acquisition (DDA) mode.

For HS-SPME-GC analysis, a TriPlus RSH SMART robotic autosampler ensured consistent pre-analytical preparation. Each 20 ml glass vial contained 5 ml of wine and 1.5 g of NaCl, sealed with a magnetic cap featuring a pierceable septum. A technical blank of bi-distilled water and a surrogate matrix blank of a 90:10 v:v water–ethanol solution were also analyzed. Samples were incubated at 40 °C for 5 min with orbital mixing at 250 rpm to achieve gas phase equilibrium. After thermal conditioning, the septum was pierced, and a DVB/Carbon WR/PDMS smart SPME fiber (80 μm thickness) was exposed 10 mm above the liquid for 30 min to extract analytes. Extraction conditions were optimized based on previous studies[72,73]. Following extraction, the fiber was retracted and transferred to the gas chromatograph's injection port, where it was thermally desorbed at 250 °C for 3 min. The SPME fiber was thermally cleaned at 240 °C for 5 min between analyses. GC–MS analysis utilized a TRACE GC 1610 Series (Thermo Fisher Scientific, Waltham, MA, USA) gas chromatograph interfaced with an Orbitrap Exploris MS (Thermo Fisher Scientific,

Waltham, MA, USA) analyzer, operating in electron impact mode (70 eV). The capillary GC column was TG-5MS (30 m ×' 0.25 mm ID, 0.15 μm film thickness), with helium as the carrier gas at a flow rate of 1.2 ml/min. The temperature program started at 40 °C (5 min hold), and ramped at 7 °C/min to 250 °C (5 min hold), totaling 40 min. The transfer line and ionization source temperatures were maintained at 270 °C, and the GC operated in split mode with a 1:20 split ratio. Mass spectra were recorded in full scan mode (50–700 Da) to collect total ion current chromatograms.

For LC-HRMSMS metabolome analysis, 1 ml aliquots of liquid wine were taken from commercial bottles after vigorous shaking. Interpooled quality control (QC) samples were created by pooling equal aliquots (200 μl) from each sample and underwent the same preparation as experimental samples. Wine aliquots and QCs were vortex-mixed for 30 s and sonicated for 10 min in a sonicator bath. After sonication, samples were transferred to Spin-X Centrifuge Tube Filters (0.22 μm, cellulose acetate membrane) and centrifuged at 14,000 rpm for 10 min. The filtered extracts were collected in glass microvials for analysis. To minimize bias, all experimental samples were randomized before the analytical run. Additionally, a QC sample was injected repeatedly (10 times) prior to the first sample injection for system equilibration and conditioning. Sample analyses were conducted in Data Dependent Acquisition (DDA) mode. LC–MS analysis was performed using an Eksigent M5 MicroLC system (Sciex) coupled with a TripleTOF 6600+ mass spectrometer featuring an OptiFlow Turbo V Ion Source (Sciex). Analyses were conducted in positive ionization mode with the column temperature set at 35 °C. A 5 μl sample volume was loaded onto a Phenomenex Luna Omega Polar C18 column (100 × 1.0 mm ID, 1.6 μm, 100 Å). Chromatographic separation occurred over 25 min at a constant flow rate of 30 μl/min, following this gradient elution program: 0–2 min, 0.2% eluent B; 2–5 min, 0.2-15% eluent B; 5–15 min, 15–70% eluent B; 15–18 min, 70–98% eluent B; 18–20 min, 98% eluent B; 20–22 min, 98–0.2% eluent B; 22–25 min, 0.2% eluent B. Equilibration time between chromatographic runs was 3 minutes. Mobile phase A consisted of 0.1% formic acid, while mobile phase B was acetonitrile with 0.1% formic acid. IonSpray voltage (ISV) was set to 5000 V, and the curtain gas supply pressure (CUR) was 30 PSI. Nebulizer and heater gas pressures were set at 30 and 40 PSI, respectively, with the ion spray probe temperature at 300 °C. The declustering potential was 80 V, and analyses were performed using a collision energy of 40 eV.

**Statistics and reproducibility**. Each subsection of the "Methods" contains detailed explanations of statistical approaches used in this paper. In brief, all statistical analyses were conducted using the R software (R Core Team; www.r-project.org, accessed March 2021), version 4.1.2, with the packages "Made4"[65], "vegan" (https://cran.r-project.org/web/packages/vegan/index.html, version 2.6-6.1), and "heatmap3"[66]. A permutation test with a pseudo-*F* ratio (function "adonis" in the vegan package) was employed to assess the suitability of data separation in the PCoA. The Wilcoxon rank-sum test and Kruskal–Wallis test were employed to ascertain whether there were significant discrepancies in relative taxon abundance between the various groups. *P*-values were corrected for multiple testing using the Benjamini–Hochberg method, with a false discovery rate (FDR) of ≤0.05 considered statistically significant. Procrustes test was employed to compare microbiomes across time points, with consideration given to the influence of seasonality and management practices (including agronomical practices and management, vine clone type, rootstock family, altitude, and soil composition). A Procrustes correlation test on the beta distribution was also performed to assess the soil-rhizospheric continuity of microbial community composition. Variations in wine metabolites related to the microbiome configurations were estimated by correlation analysis and calculated using the "envfit" function in the vegan R package.

## Reporting summary

Further information on research design is available in the Nature Portfolio Reporting Summary linked to this article.

## Data availability

Both metagenomics and metabarcoding data generated during the current study are available in the ENA archive under the accession number PRJEB75007. Please refer to the 'library_name' column in the metadata to distinguish between shotgun, ITS, or 16rRNA sequencing sources. Source data for the graphs presented in Fig. 3 can be found in the Supplementary Data 1.

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

## Acknowledgements

This work was carried out in the context of the "Controlling Microbiomes Circulations for Better Food Systems" (CIRCLES) project, which was funded by the European Union's Horizon 2020 research and innovation program under grant agreement no. 818290. We gratefully acknowledge the Consorzio del Vino Nobile di Montepulciano for providing access to vineyards and their invaluable assistance during sampling campaigns and covariates collection. Their cooperation significantly contributed to the success of this study.

## Author contributions

M.C. and S.R.: conceptualization. G.P., E.N., D.S. N.I., J.F., and S.R.: data curation, formal analysis. M.C. and S.R.: project administration, resources, supervision. G.P., E.N., D.S., N.I., and S.R.: visualization. G.P. and S.R.: writing—original draft. E.N., D.S., N.C., L.F., A.C., A.G.V.R., N.I., J.F., S.T., and M.C.: writing—review and editing.

## Competing interests

The authors declare no competing interests.
