## [Transparent Peer Review file · Communications Biology]

Zonation of the *Vitis vinifera* microbiome in Vino Nobile di Montepulciano PDO production area

Corresponding Author: Dr Simone Rampelli

Version 0:

Reviewer comments:

Reviewer #1

(Remarks to the Author)

The authors describe microbial diversity of rhizosphere and soil samples from sub-regions within a PDO area. This is an exciting dataset collected from Italian vineyards, however I have several concerns with major methodological issues that require substantial alterations to the manuscript and the data analysis.

Some key issues:

1. Key Methodology missing: It is not described how the authors compared their own dataset to the global analysis of Gobbi et al (2022). For an adequate comparison the raw data of both studies would need to be analysed together as bioinformatic choices have a huge impact on the results. Additionally different molecular biological methods were used, e.g. different primers (Gobbi et al. used 515/806 for 16S and ITS1F/ITS2 for ITS, while here the authors use 341F/785R for 16S and ITS3/ITS4 for ITS) as well as DNA extraction methods (Dneasy Powerlyzer Powersoil Kit in Gobbi et al. and DNAesy PowerSoil Pro Kit here), which further limits comparability. The authors should acknowledge and discuss these limitations in their work, as it significantly limits the relevance to directly compare these studies.
2. A large part of the introduction refers to the effects of terroir on wine characteristics, yet in this study no wine characteristics are studied. If this should still be the main focus of the study, the authors should make the link between the microbial communities in the soil and rhizosphere and the resulting organoleptic properties clearer (e.g. Line 76).
3. There are substantial confounding factors between the different vineyards, e.g. for different management practices, rootstocks, chemical soil properties etc., which will directly influence present soil and rhizosphere communities. The authors therefore must acknowledge, test for and discuss these covariates.
4. A central argument of the authors is that there is a need for a finer scale analysis of vineyard microbiomes (e.g. Line 66). However the distance between the outer regions (e.g. San Biagio and Valiano) is around 15km, while previous studies have also focused on fine-scale analysis of specific viticultural zones, including vineyards in closer proximity, neighboring sites (e.g., Burns 2015, Bokulich 2016), or even intra-vineyard variation (e.g., Setati et al. 2012).
5. The conclusions appear to be drawn out of proportion, e.g. the study is not "showing the importance of local microbial diversity in maintaining plant health and productivity and potentially wine quality" (Line 334) as these outcomes are not assessed. Further it is claimed that this study 'represents the first steps towards new strategies [...] for the protection and preservation of local microbial terroir features' (Line 341, and similarly in 346, 354 29, 44), which is also not justified since the effects of these varying microbial communities are not studied, not how they could be 'preserved'.

If these issues are addressed, this paper will be a useful addition to the literature on microbial terroir in viticulture.

Individual Line Comments

29 What is specifically meant with protecting microbial terroir?

34 Microbial terroir describes regional strain variety that may contribute to wine characteristics. To demonstrate that a region has a characteristic microbial terroir would therefore require an assessment of the resulting wine characteristics and the influence of distinct microbial communities thereon.

36 I would kindly ask the authors to be more precise in their formulations and word choices as these can easily lead to wrong statements. e.g. core taxa provide PGP traits.

60 What is meant here with 'vineyard microbiome components' - vineyard microbial communities?

69 The whole introduction to this point was about how terroir is reflected in wine qualities, yet the entire research is based on rhizosphere and soil samples - please make the connection between how the soil microbiome may have an influence on terroir more explicit.

72 Why is this a delicate balance? This is not described in the Results.

74 How is this study then connecting the soil microbiome to distinct organoleptic properties?

- 86 Are there better sources on this than doctorwine.it and quoting 'local wine merchants'?
- 120 To assess spatial variation please apply appropriate statistical tests, e.g. to assess effects of geographical distance in community composition, consider applying a Mantel test with the geodesic distance between samples.
- 129 Differential abundance testing should be applied to assess variation in abundance between regions.
- 131 What is the reason for this analysis being performed on a genus level instead of directly on the ASVs?
- 150 How was a continuity between soil and rhizosphere microbiome shown? I suggest e.g. differential abundance testing or simply comparing the beta diversity between the groups to truly test the taxonomic differences. Simply reviewing the lists of core genera is not sufficient.
- 154 This could be shown with a Mantel test as described above.
- 158 Did you test the effect of these confounding factors like agronomical practices, rootstock etc. for example with ADONIS or PERMANOVA? Please show some statistics for this instead of just visually interpreting PCA plots.
- 163 There is quite a lot of methodological description missing, e.g. here it is mentioned that Procrustes analysis was performed but this has not been described anywhere.
- 166 Kruskal Wallis tests should not be run on relative abundance as leads to a high false positive rate. Please test for differential abundance instead of using relative abundance for statistical testing.
- 167 There are 14 different vineyards in the 12 AUGs were sampled but the intra-AUG diversity is not discussed. Also, how was the data from the two vineyards within a single AUG combined?
- 169 How were these significantly discriminating genera retrieved? And why was relative abundance tested afterward. I suggest testing differential abundance of all ASVs/genera and recreating the heatmap thereof.
- 212 What about fungal MAGs?
- 289 What testing was performed to show variation with the season? It's great that samples were taken at 4 different time points but there was no longitudinal analysis or comparison of time points.
- 291 The testing for the correlation with pedology was not described in the Results?
- 305 Where was this homogeneity shown? Was there a test performed for this?
- 314 How was this shown - that the different AUGs have a greater potential for e.g. P solubilization?
- 334 How do you show the importance of diversity? The diversity is reported but there are no causal connections drawn in this study.
- 336 I believe climate change is meant here instead of global change. I would also argue that there are manifold contributions to potential loss of microbial diversity.
- 428 The ENA accession number is not valid yet.
- 763-767 the URLs are not cited correctly.

Supplementary Table 1: it is fantastic that all this vineyard metadata on different management practices, different clones and on different rootstocks was collected! However, these are all confounding factors which must be discussed and analyzed accordingly. Chemical soil properties are from different AGUs (12) not vineyards (14) → how were these evaluated? And if they are not from the specific vineyards justify this as a confounding factor?

Reviewer #2

(Remarks to the Author)

Palladino et al. present a manuscript that includes a very large set of different kind of data. There is a metabarcoding analysis of both fungi and bacteria from both bulk soil and rhizosphere; in addition, a subset of samples was used for shotgun metagenomics analysis. The samples investigated originated from 12 different areas of production of a specific Italian wine, all confining and located in the same territory. At the beginning (also from the title), it appears that the aim of the work was to discriminate the microbial terroir across these 12 areas of production; however, when going on with reading the manuscript, the results presented expand behind this aspect. The entire territory is compared to the rest of the world, but only considering the bulk soil; then the 12 areas are compared, but only considering the rhizospheric soil. I wonder whether the experimental conditions are so similar to allow such a broad comparison, considering that all the existing data used originate from a different survey.

So, at the end, the work presented is very confusing and the reader needs to decipher all the aspects that should be well clarified by the authors instead. Several aspects are not clear and the results are not presented explicitly (see specific comments and questions below).

Moreover, several details about the sampling strategy and the experimental design are missed, and this prevents the possibility to evaluate the reliability of the results.

Specific comments and questions:

- L. 21: *Vitis vinifera* in italics.
- L. 28: a space is missed after the second comma.
- L. 69: it is not clear what the authors means with "the interplay between bulk and rhizospheric soil microbiomes". To my understanding, this study is about the plant-soil interaction at root level. In this kind of studies, the bulk soil is a sort of control, used to elucidate the filtering role of roots in selecting specific microbial populations. However, this seems to not be the case here, but the authors should clarify better their concepts and ideas (rationale) behind the work then.
- L. 84: replace "each of which has" with "each of them showing"
- L. 86: "Local wine merchants claim that..." Is there any scientific evidence of this? For example, can the authors indicate any sensorial test that was done under reliable and statistically robust conditions? It is obvious that every wine producer will firmly state that its wine is unique, but that is not science, it's called marketing!
- L. 102-103: this information should be moved (or, at least, repeated) in the materials and methods.
- L. 110: why the fungal sequenced samples are so much less than the bacterial ones?

- In all figures, the scientific names must be written in italics.
- L. 158-160: this statement is not clear. The authors should clearly indicate whether the additional variables (rootstocks, agronomical practices, soil composition, etc.) are statistically significant factors that discriminate the sample groups.
- L. 200: remove one dot.
- L. 362-364: the sampling strategy and the experimental design must be better described. It is not stated how many samples were taken per each site, the number of replicates, how each sample was obtained (how many plants/soil cores contributed to one sample), and so on.
- L. 417: how did you select these 14 subsamples?
- L. 438: to my knowledge, the VSEARCH tool is used to detect chimeras, not to assign taxonomy.
- L. 441: what about mitochondrial and chloroplast sequences? Didn't you consider them?

Reviewer #3

(Remarks to the Author)

In this study, the authors define the microbial terroir of vineyards in the "Consorzio del Vino Nobile di Montepulciano DOCG" PDO area in Italy. Using rhizospheres and soil samples, the study reveals microbiomes are "AGU-specific" in terms of taxonomic abundances and plant growth-promoting functions. The study emphasizes the importance of protecting microbial terroir and biodiversity for high-quality traditional wine production. In general, the result is interesting and important. However, more analysis needs to be done to strengthen the MS. A thorough review of the manuscript revealed the following observations.

1. The author is encouraged to highlight the novelty of this study, particularly in the context of existing literature on wine terroir. The integration of microbial composition and functional profiling with a longitudinal study and comprehensive metadata represents a unique and innovative approach.
2. Line 385 - Consider clarifying if "ITS analysis" refers to fungal community analysis.
3. Line 440 - Has the author conducted an analysis of the archaeal community? Given that the V3-V4 primer may amplify archaeal fragments, it is recommended to consider using the term "prokaryotes" instead of "bacteria" throughout the manuscript if the archaeal group is not excluded.
4. Line 462 - Please elaborate on the methods used to remove low-quality reads and reads originating from the host.
5. Line 431 - Has the data been normalized prior to conducting alpha and beta diversity analyses?
6. Line 465 - How were redundant genomes handled, particularly in instances where multiple metagenome-assembled genomes (MAGs) may be identical? Were non-redundant MAGs created for further investigation into MAG function?
7. In the Results section, consider removing repetitive information that has already been discussed in the Materials and Methods section (e.g., lines 188 and 224).
8. The interpretation of results at lines 228 and 300 may benefit from clearer implications. Consider exploring whether taxa with plant growth promotion traits were enriched in the rhizosphere due to interactions with host plants.
9. Is the absence of certain genes attributed to their true absence or incomplete genome assemblies?
10. The unique functional profiles of microbial plant growth-promoting traits observed in the rhizosphere microbiomes of different agricultural units (AGUs) should be further linked to environmental factors such as the presence of phosphorus. Consider investigating if the enrichment of bacteria with phosphorus-solubilizing capabilities is influenced by varying phosphorus availability in different regions.
11. Figure 2 shows that soil microbial communities from different countries do not strictly follow pedoclimatic zones. Investigate the potential reasons for the observed clustering in Figure 1.
12. Line 289 - The absence of a sampling season effect on grapevine rhizosphere communities is intriguing. Explore further whether the grapevine maintains a stable rhizosphere community regardless of seasonal fluctuations.
13. Additional considerations and revisions may be needed based on the above feedback for enhanced clarity and depth in the manuscript.
14. This reviewer is missing information regarding the diversity of the microbial community? Are there differences in bacterial diversity between the sites? Why this part is not done. It would be interesting to investigate the microbial richness and diversity between the locations.
15. The authors have a very nice data that could be explored more – for instance, is there any association between the bacterial and fungal community?

Version 1:

Reviewer comments:

Reviewer #1

(Remarks to the Author)

I would like to thank the authors for their thoughtful revisions as they addressed all of my comments and questions. The manuscript is now much clearer, and the extended methodological descriptions, along with the additional statistical tests, greatly enhance the rigor of the study. Additionally, it is commendable that the authors included further metabolomics analyses, which provide valuable insights. I recommend publication of this manuscript and believe that it will be a useful resource for the field.

Reviewer #2

(Remarks to the Author)

The authors satisfactorily addressed my previous comments. Congratulation for the nice work.

Reviewer #3

(Remarks to the Author)

The authors have put in significant effort and have thoroughly addressed all of my comments. I am pleased to support the publication of this paper.

Reviewer #1 (Remarks to the Author):

The authors describe microbial diversity of rhizosphere and soil samples from sub-regions within a PDO area. This is an exciting dataset collected from Italian vineyards, however I have several concerns with major methodological issues that require substantial alterations to the manuscript and the data analysis.

Some key issues:

1. Key Methodology missing: It is not described how the authors compared their own dataset to the global analysis of Gobbi et al (2022). For an adequate comparison the raw data of both studies would need to be analyzed together as bioinformatic choices have a huge impact on the results. Additionally different molecular biological methods were used, e.g. different primers (Gobbi et al. used 515/806 for 16S and ITS1F/ITS2 for ITS, while here the authors use 341F/785R for 16S and ITS3/ITS4 for ITS) as well as DNA extraction methods (Dneasy Powerlyzer Powersoil Kit in Gobbi et al. and DNAesy PowerSoil Pro Kit here), which further limits comparability. The authors should acknowledge and discuss these limitations in their work, as it significantly limits the relevance to directly compare these studies.

We thank the Reviewer for pointing out this incongruence. In order to avoid bias due to ASVs assignment, given the different primers used, we compared Gobbi's and our dataset at the genus levels, but not at the ASVs level. Moreover, Gobbi's raw sequences downloaded from public datasets were re-analyze by the same bioinformatic pipeline used for our dataset. We now specified this information in the Methods section, lines 491-498. For what concern biological methods, despite the DNA extraction methods were different between Gobbi's and our study, they are both well-established methods for DNA extraction of soil samples, thus we hope this explanation would be satisfactory for the Reviewer to address their concerns. Finally, we employed the ADONIS test to ascertain whether there were any discernible biases attributable to the "study effect" directly affecting the outcomes of the PCoA analyses, both for 16S rRNA and ITS sequencing. The p-values for these comparisons are greater than 0.05. It is our hope that this evidence will serve to dispel any remaining doubts.

2. A large part of the introduction refers to the effects of terroir on wine characteristics, yet in this study no wine characteristics are studied. If this should still be the main focus of the study, the authors should make the link between the microbial communities in the soil and rhizosphere and the resulting organoleptic properties clearer (e.g. Line 76).

I am grateful to the Reviewer for their insightful comment, which has provided us with the valuable opportunity to further elucidate our findings. This revised manuscript incorporates a metabolic analysis of the wines from the 2022 vintage, specifically those produced from grapes sourced exclusively from a single AGU. Our findings suggest a potential correlation between the microbial configurations of the AGUs and wine metabolites, which may reflect the geographical distribution (p-value = 0.04, Procrustes test). In particular, we found a significant association between the variation in concentration of five molecules and the variation in rhizosphere microbiome among the different AGUs. These include L-acetylcarnitine, L-methionine, quercitrin, citicoline and adenine. It is noteworthy that the concentration of the first three compounds has been demonstrated to exert a discernible influence on the organoleptic properties of wine. Further details and references can be found in the results and discussion section of this version of the manuscript (lines 265-282 and 356-373).

3. There are substantial confounding factors between the different vineyards, e.g. for different management practices, rootstocks, chemical soil proper.es etc., which will directly influence present soil and rhizosphere communities. The authors therefore must acknowledge, test for and discuss these covariates.

We agree with the Reviewer that confounding factors between the different vineyard should be acknowledged and taken into account when discussing terroir microbiome geographical segregation. This is why we applied a permutational test to determine the spatial distances determined by the dissimilarity of rhizospheric microbial communities in different vineyards, also taking into account the aforementioned confounding factors (seasonality, agronomical practices and management, vine clone type, rootstock family, altitude, and soil composition). We included this information in Figure 4 description, and we also slightly modify the text in the Results section, lines 163-164. We hope that this will increase clarity on the confounding factors, addressing the Reviewer's concerns.

4. A central argument of the authors is that there is a need for a finer scale analysis of vineyard microbiomes (e.g. Line 66). However the distance between the outer regions (e.g. San Biagio and Valiano) is around 15km, while previous studies have also focused on fine-scale analysis of specific viticultural zones, including vineyards in closer proximity, neighboring sites (e.g., Burns 2015, Bokulich 2016), or even intra-vineyard variation (e.g., Setati et al. 2012).

We thank the Reviewer for their valuable comment. In the sentences to which they refer, we have used as a comparative term the work of Gobbi and colleagues (2022), in which the authors describe, with unprecedented effort, how the microbial configurations of the vineyard reflect geographical distances at both the country and regional levels. The Reviewer is therefore right to point out that there are other works that have focused on microbiome variation at a more detailed scale, even characterizing microbiome variation within the same vineyard, however here we focus more on the terroir scale, so PDO and its local declination at AGUs. Our purpose was to provide some glimpses on microbiome features associated with the territory, at the finer AGU scale, from which variants of the Montepulciano wine production are historically obtained. Thus, with our work we would like to highlight that it is necessary to perform a fine-scale analysis (with reference to Gobbi and colleagues) at the PDO area (or sub-area) level, i.e. focusing at the level of the territory uniquely associated with a wine, in order to capture the scale of microbiome variation that is of interest for wine production (in our case two levels: the entire territory of Vino Nobile di Montepulciano and the AGUs). We hope that this explanation has clarified our perspective.

5. The conclusions appear to be drawn are out of proportion, e.g. the study is not "showing the importance of local microbial diversity in maintaining plant health and productivity and potentially wine quality" (Line 334) as these outcomes are not assessed. Further it is claimed that this study 'represents the first steps towards new strategies [...] for the protection and preservation of local microbial terroir features' (Line 341, and similarly in 346, 354 29, 44), which is also not justified since the effects of these varying microbial communities are not studied, not how they could be 'preserved'.

If these issues are addressed, this paper will be a useful addition to the literature on microbial terroir in viticulture.

The Reviewer is right, and we thank them for raising these issues. On the one hand, we believe that our results showed that many bacteria at the soil-plant interface have plant growth-promoting functions that could be beneficial for plant health. On the other hand, we understand that our statements may seem speculative. For this reason, in this new version of the manuscript, we provide a new wine metabolomic analysis to connect the terroir microbiome configurations to the quality of the final product. To this aim we metabolically analyzed the wines of the 2022 vintage (that of the sampling campaigns) in those cases where the wine was produced from grapes taken exclusively from a single vineyard (AGU). We found that there are correlations between the microbial configurations of the AGUs and wine metabolites that reflect the geographical distribution (p-value = 0.04, Procrustes test, see new Supplementary Figure 3 and text within the Results and

Discussion sections, (lines 265-282 and 356-373). We believe that this new analysis can experimentally justify what we wrote in the first version of the manuscript. Anyway, we wanted to tone down the statements highlighted in the reviewer's comment so that they appear less speculative (lines 28-30, 374-375, 378-382, 385-387).

We absolutely agree with the Reviewer that this work can pave the way to more research and future microbiome applications, and we are already working on it. For instance, > 40 plant growth promoting isolates have been obtained and genomic sequenced from the Montepulciano PDO territory, and AGUs specific plant growth promoting species have been obtained. We are now cultivating these taxa for in vitro tests, first to assess for plant growth promoting functions, then for testing, in pot, the plant growth promoting activity, before running field test. All this work, now in progress, is part of the CIRCLES EU project, with the ultimate goal to provide the AGUs with tailored bio-promoters, to preserve (and protect) the key microbiome determinants of the soil microbiome features.

We hope this makes the manuscript complete and more acceptable.

Individual Line Comments

29 What is specifically meant with protecting microbial terroir?

We believe that characterization and preservation of the local peculiarities of the microbial terroir in the different AGUs is a key factor to safeguard the productions of high-quality traditional wines. For instance, if we assume that the local specificities of the microbiome terroir are somehow part of the production process, as regulating the plant health, physiology and biochemistry, we need to protect these natural microbiome features, particularly in the context of the current global changes, that are threatening the global microbiomes (Cavicchioli et al., 2019). We implemented this sentence in lines 28-30 to improve clarity.

34 Microbial terroir describes regional strain variety that may contribute to wine characteristics. To demonstrate that a region has a characteristic microbial terroir would therefore require an assessment of the resulting wine characteristics and the influence of distinct microbial communities thereon.

We are grateful to the Reviewer for their insightful comment, which enabled us to conduct a more thorough examination of a crucial element of our article. The revised version incorporates metabolomic analysis of the wine from the identical 2022 vintage in which the sampling was conducted, encompassing those wineries that produced the wine using grapes from the vineyards of a single AGU. As stated above in a previous concern, a correlation was identified between the wine metabolic profile and the rhizosphere microbiome configuration of the correspondent AGUs, with some metabolites present at varying concentrations in the analyzed wines (lines 265-282 and 356-373). We are confident that this analysis will provide sufficient resolution to address the concerns raised by the Reviewer.

36 I would kindly ask the authors to be more precise in their formulations and word choices as these can easily lead to wrong statements. e.g. core taxa provide PGP traits.

We thank the Reviewer for the suggestion, and we modified the text accordingly (lines 36-37).

60 What is meant here with 'vineyard microbiome components' - vineyard microbial communities?

Yes, we mean the microbiome communities characterizing the vineyards terroir. We modified the sentence as suggested by the Reviewer (line 60).

69 The whole introduction to this point was about how terroir is reflected in wine qualities, yet the entire research is based on rhizosphere and soil samples - please make the connection between how the soil microbiome may have an influence on terroir more explicit.

We thank the Reviewer for their comment, and we agree that the interconnection between soil microbiome and terroir should be made more explicit, in order to justify its possible relevance in

defining distinct organoleptic characteristics of wines of different areas. We implemented this connection in lines 71-74 to improve clarity.

72 Why is this a delicate balance? This is not described in the Results.

The Reviewer is right. We changed the sentence by removing "delicate balance" (lines 70-72).

74 How is this study then connecting the soil microbiome to distinct organoleptic properties?

In this new version of the manuscript, we implemented a new metabolomic analyses of the wines from the 2022 vintage in cases where the wine was produced from grapes exclusively from a single vineyard (AGU). Thanks to this new analysis, we provide evidence of associations between differences in terroir and differences in the composition of wine metabolites. Even if this is not a causal relationship, it can help in the understanding of how natural biological processes, including the relation microbial terroir-wine, can influence the final product. We implemented this description in the Results and Discussion session at lines 265-282 and 356-373.

86 Are there better sources on this than doctorwine.it and quoting 'local wine merchants'?

The Reviewer is correct in their assessment that the initial description was insufficiently clear. The revised text (lines 86-89) provides a more accurate and detailed account of the intended meaning. Furthermore, the revised manuscript includes a new metabolomic analysis as described right above to support our statement that wines exhibited different organoleptic profiles which reflected the specific characteristics of the terroir. We hope this clarification will be sufficient for the Reviewer.

120 To assess spatial variation please apply appropriate statistical tests, e.g. to assess effects of geographical distance in community composition, consider applying a Mantel test with the geodesic distance between samples.

We thank the Reviewer for this comment. We used ADONIS ("adonis2" function of the "vegan" R package), which is a permutation test with pseudo-F ratio, considered the gold standard in microbial ecology for this type of analysis.

129 Differential abundance testing should be applied to assess variation in abundance between regions.

The Kruskal-Wallis test is widely used in microbial ecology to compare the relative abundances of taxa in more than two groups, as reported in these papers by other researchers on microbiome science (not only the microbiome of vineyards, but also of other hosts studied more extensively by the scientific community: e.g. ref1, ref2, ref3). Therefore, we believe that our statistical approach is consistent with that of other studies in the field and we do not believe that there is a need to apply different tests. However, we ask the reviewer if we have misinterpreted his comment, we are more than willing to intervene on the manuscript.

131 What is the reason for this analysis being performed on a genus level instead of directly on the ASVs?

As per Reviewer previously suggestion, there are differences in the raw data production between Gobbi's and our study, due to different DNA regions being sequenced for bacterial and fungal characterization. In order to avoid bias due to ASVs assignment, we compared Gobbi's and our dataset at genus level, but not at ASVs level. We now specified this information in the Material and Methods section, lines 491-498, and we hope this clarification will be sufficient for the Reviewer as per why we conducted this analysis at the genus level instead of directly on the ASVs.

150 How was a continuity between soil and rhizosphere microbiome shown? I suggest e.g. differential abundance testing or simply comparing the beta diversity between the groups to truly test the taxonomic differences. Simply reviewing the lists of core genera is not sufficient.

As per Reviewer's suggestion, we performed a Procrustes correlation test using `protest` function in R to compare the beta diversity distribution of soil and rhizospheric samples. We added in the text at lines 147-150 and 512-516 the p-value for the performed test.

154 This could be shown with a Mantel test as described above.

We used ADONIS (“adonis2” of the “vegan” R package), as explained above.

158 Did you test the effect of these confounding factors like agronomical practices, rootstock etc. for example with ADONIS or PERMANOVA? Please show some statistics for this instead of just visually interpreting PCA plots.

We thank the Reviewer for pointing this out. We performed indeed a permutation test with pseudo-F ratio on the different groups resulting from the PCoA plot, testing the segregation pattern also for the aforementioned confounding factors (seasonality, agronomical practices and management, vine clone type, rootstock family, altitude, and soil composition). We included this information in Figure 4 description, and we also slightly modify the text in the Results section, lines 163-164. We hope that this will increase clarity on the confounding factors, addressing the Reviewer’s concerns.

163 There is quite a lot of methodological description missing, e.g. here it is mentioned that Procrustes analysis was performed but this has not been described anywhere.

We apologize for the lack of clarity on this point. We implemented the methodology section in lines 491-498 and 507-509. We hope that this will implement this section enough for the Reviewer.

166 Kruskal Wallis tests should not be run on relative abundance as leads to a high false positive rate. Please test for differential abundance instead of using relative abundance for statistical testing.

We thank the Reviewer for the comment. To the best of our knowledge, the Kruskal-Wallis rank sum test is a classical statistical test included in differential abundance approaches (DA). As mentioned in a previous response, the method is widely used with relative abundances in microbiome science. We add that in a recent study where different DAs are applied, Kruskal-Wallis is used with relative abundances without highlighting biases compared to the other DAs (ref). Therefore, we believe that our statistical approach is consistent with approaches commonly used in microbiome science and should not be modified. Furthermore, it should be noted that the P values are always corrected using the Benjamini–Hochberg method, which serves to minimise the rate of false discovery.

167 There are 14 different vineyards in the 12 AUGs were sampled but the intra-AUG diversity is not discussed. Also, how was the data from the two vineyards within a single AUG combined? The observation raised by the Reviewer is intriguing, however, it is beyond the scope of our current work, which has focused on the diversity between AGUs, and has revealed that this is in fact higher than that observed within AGUs. Indeed, we found that rhizosphere microbiome separation in the PCoA correlated with geographic separation in terms of distance (in meters) between vineyards (ADONIS, (p-value ≤ 0.003 , line 169). This observation led us to conclude that, in our case, the intra-AGU variability was smaller compared to the inter-AGU variability.

169 How were these significantly discriminating genera retrieved? And why was relative abundance tested afterward. I suggest testing differential abundance of all ASVs/genera and recreating the heatmap thereof.

We apologize to the reviewer for the lack of clarity. We used RandomForest, combined with the Kruskal-Wallis test among relative taxon abundances in each AGU, to identify the discriminating genera in the different AGUs, and then plotted the results as a heatmap using their relative abundance in each AGU. We have changed the text accordingly to make it clearer (lines 172-176).

212 What about fungal MAGs?

We reconstructed MAGs from shotgun metagenomics data. Some were assigned to eukaryotic microorganisms but had very low completeness and contamination parameters and were therefore not considered in this work.

289 What testing was performed to show variation with the season? It’s great that samples were taken at 4 different time points but there was no longitudinal analysis or comparison of time points. We found that the geographical segregation of the AGUs microbiome was robust not only to the confounding factors, as described above, but also to seasonality (Procrustes test, p-value ≤ 0.01),

leading us to think that the main factor driving microbiome differentiation was geographical origin at a very local scale, rather than plant maturity and season. We described this test in lines 164-166 and this is why we decided not to discuss longitudinal analysis at specific timepoints. We hope that the Reviewer agrees with our choice.

291 The testing for the correlation with pedology was not described in the Results?

The reviewer made a very fair point. Soil composition data were among the confounding factors included in our PCoA analysis to evaluate the difference in microbial composition of rhizospheres in various vineyards (Figure 4). We hope this will clarify this issue.

305 Where was this homogeneity shown? Was there a test performed for this?

We are grateful to the reviewer for bringing this point to our attention. We have now clarified that we were referring to the fact that genes coding for these functions were very widespread in the genomes analysed. We hope that this has been adequately addressed in the new version of the manuscript.

314 How was this shown - that the different AUGs have a greater potential for e.g. P solubilization?

We apologize for the lack of clarity on this point. We showed in Figure 7 that the AGUs of the southeastern part of the production area (the side delimited by Argiano) showed a greater potential for P solubilization, while those of the western part (delimited by S.Biagio) showed a greater propensity for ACC deaminase production. We conducted Wilcoxon tests on the read-mapping results, comparing the values in the AGUs to the south-east (e.g. Argiano, Cervognano) with those to the west (e.g. S. Biagio). The results indicated a significant difference (p -value = 0.05), which we believe is worthy of mentioning. These findings were also supported by the fact that the MAG assigned to *Conexibacter*, associated with the southeastern part of the territory, encoded genes devoted to P solubilization, while the MAG assigned to *Streptomyces*, associated with the western and southern parts of the territory, carried the ACC deaminase gene (Figure 8 and supplementary Table 5). We hope that this will clarify the Reviewer's concerns.

334 How do you show the importance of diversity? The diversity is reported but there are no causal connections drawn in this study.

We show that the Montepulciano territory compared to the rest of the world, and within the Montepulciano territory, each AGU compared to another, has a unique microbiological fingerprint that encodes functions that favor plant growth. This is important for us and underlines how wine is produced in a unique territory, not only from an environmental/pedoclimatic point of view, but also from a microbiological point of view. Furthermore, as the Reviewer mentioned that there are not present causal relationships, in this new version of the manuscript, thanks to the new metabolomic analysis of the wines, we provide evidence of associations between differences in terroir and differences in the composition of wine metabolites (lines 265-282 and 356-373). Even if this is not a causal relationship, because it depends on various factors within the terroir (please note that VINO Nobile di Montepulciano is produced with very strict procedural guidelines, i.e. everyone who wants to make wine makes it the same way), it is a small step towards understanding how natural biological processes, including the relation microbial terroir-wine, can influence the final product.

336 I believe climate change is meant here instead of global change. I would also argue that there are manifold contributions to potential loss of microbial diversity.

We would like to extend our sincerest apologies to the Reviewer for any confusion. To clarify, we are referring to global changes, not climate changes. The term "climate change" is typically used to describe long-term shifts in temperatures and weather patterns. "Global change," on the other hand, encompasses all changes within the Earth system, not just temperatures and weather. These changes can impact the atmosphere, water and soil, and land. If the Reviewer agrees, we believe that "global changes" might be a more suitable term to use in our manuscript.

428 The ENA accession number is not valid yet.

We apologize for this. However, we decided to not make the sequences public before the article publication. In case the reviewer would like to access the sequences, please let us know so we can release the study on ENA and make them accessible.

763-767 the URLs are not cited correctly.

We apologize but we could not understand the exact problem with the sitography because we could not find any precise information in citing in the journal guidelines. However, we dedicated a new section ("Sitography", line 905) to the cited URLs, we inserted the last access date, where applicable, both in the text and in the Sitography section (e.g. lines 517 and 910-915), we inserted the tools version where applicable (e.g. line 507), and we inserted the title for PDF documents linked to the URL (e.g. lines 906-907). We hope this will be sufficient to correct the URLs citations, however we remain available for any other modification.

Supplementary Table 1: it is fantastic that all this vineyard metadata on different management practices, different clones and on different rootstocks was collected! However, these are all confounding factors which must be discussed and analyzed accordingly. Chemical soil properties are from different AGUs (12) not vineyards (14) → how were these evaluated? And if they are not from the specific vineyards justify this as a confounding factor?

We thank the Reviewer for this appreciation, and we agree with the Reviewer that confounding factors between the different vineyard should be acknowledged and taken into account when discussing terroir microbiome geographical segregation. This is why we applied a permutational test to determine the spatial distances determined by the dissimilarity of rhizospheric microbial communities in different vineyards, also taking into account the aforementioned confounding factors (seasonality, agronomical practices and management, vine clone type, rootstock family, altitude, and soil composition). We included this information in Figure 4 description, and we also slightly modify the text in the Results section, lines 163-164. This led us to conclude that the main factor driving microbiome differentiation was geographically generated at a very local scale (AGUs), even when there were multiple vineyards in the same AGU. Indeed, we found that rhizosphere microbiome separation in the PCoA correlated with geographic separation in terms of distance (in meters) between vineyards (ADONIS, (p-value \leq 0.003, line 169). Thus, in our case, the intra-AGU variability was smaller compared to the inter-AGU variability. We hope that this will increase clarity on the confounding factors, addressing the Reviewer's concerns.

Reviewer #2 (Remarks to the Author):

Palladino et al. present a manuscript that includes a very large set of different kind of data. There is a metabarcoding analysis of both fungi and bacteria from both bulk soil and rhizosphere; in addition, a subset of samples was used for shotgun metagenomics analysis. The samples investigated originated from 12 different areas of production of a specific Italian wine, all confining and located in the same territory. At the beginning (also from the title), it appears that the aim of the work was to discriminate the microbial terroir across these 12 areas of production; however, when going on with reading the manuscript, the results presented expand behind this aspect. The entire territory is compared to the rest of the world, but only considering the bulk soil; then the 12 areas are compared, but only considering the rhizospheric soil. I wonder whether the experimental conditions are so similar to allow such a broad comparison, considering that all the existing data used originate from a different survey.

So, at the end, the work presented is very confusing and the reader needs to decipher all the aspects that should be well clarified by the authors instead. Several aspects are not clear and the results are not presented explicitly (see specific comments and questions below).

Moreover, several details about the sampling strategy and the experimental design are missed, and this prevents the possibility to evaluate the reliability of the results.

We would like to thank the Reviewer for their thoughtful and constructive feedback. We have taken the Reviewer's recommendations into account and revised the manuscript accordingly. We hope that we have adequately addressed all comments to the best of our ability. Regarding the comparison with the samples from the previous study by Gobbi and colleagues (2022), we opted to utilize the soil samples because they were the only available specimens within that study. However, for a comparison of the AGUs within the Montepulciano area, we felt it would be more beneficial to focus on the rhizosphere, particularly for the analysis of plant-growth promoting functionalities, given that it represents the soil-plant exchange interface. Please refer to the sections below for responses to the other points raised in this initial comment.

Specific comments and questions:

- L. 21: *Vitis vinifera* in italics.

Modified as suggested.

- L. 28: a space is missed after the second comma.

Modified as suggested.

- L. 69: it is not clear what the authors means with “the interplay between bulk and rhizospheric soil microbiomes”. To my understanding, this study is about the plant-soil interaction at root level. In this kind of studies, the bulk soil is a sort of control, used to elucidate the filtering role of roots in selecting specific microbial populations. However, this seems to not be the case here, but the authors should clarify better their concepts and ideas (rationale) behind the work then.

We apologize for the lack of clarity on this point. In our study, we used the characterization of microbial soil samples in order to demonstrate the specificity of the Montepulciano territory compared to vineyards all around the world, and also between different AGUs in the same territory. We chose to use soil samples for this purpose because of the much easier and bigger availability in the scientific literature on the topic of microbial soil samples in other vineyards around the world. However, the rhizospheric soil is the one in very close contact with the plant, thus we should focus on it when investigating possible distinct organoleptic characteristics of wines from specific regions. After investigating the specificity of the Montepulciano territory compared with other vineyards, we demonstrated the continuity between bulk soil and rhizospheric samples (lines 147-150), thus we were able to focus on these last samples to characterize the local terroir and possible implications on the organoleptic properties of the final product. We hope this explanation will clarify the Reviewer’s concerns about this point.

- L. 84: replace “each of which has” with “each of them showing”

Modified as suggested.

- L. 86: “Local wine merchants claim that...” Is there any scientific evidence of this? For example, can the authors indicate any sensorial test that was done under reliable and statistically robust conditions? It is obvious that every wine producer will firmly state that its wine is unique, but that is not science, it's called marketing!

The Reviewer is correct in their assessment that the initial description was insufficiently clear. The revised text (lines 86-89) provides a more accurate and detailed account of the intended meaning. Furthermore, the revised manuscript incorporates a metabolic analysis of the wines from the 2022 vintage, specifically those produced from grapes sourced exclusively from a single AGU. Our findings suggest a potential correlation between the microbial configurations of the AGUs and wine metabolites, which may reflect the geographical distribution (p-value = 0.04, Procrustes test). In particular, we found a significant association between the variation in concentration of five molecules and the variation in rhizosphere microbiome among the different AGUs. These include L-acetylcarnitine, L-methionine, quercitrin, citicoline and adenine. It is noteworthy that the concentration of the first three compounds has been demonstrated to exert a discernible influence on the organoleptic properties of wine. Further details and references can be found in the results

and discussion section of this version of the manuscript (new Supplementary Figure 5 and text within the Results and Discussion sections, lines 265-282 and 356-373).

- L. 102-103: this information should be moved (or, at least, repeated) in the materials and methods. As suggested by the Reviewer, we repeated this information in the material and methods section (lines 407-408).

- L. 110: why the fungal sequenced samples are so much less than the bacterial ones? The ITS analysis was conducted on a subset of the collected samples, ensuring their representativeness (at least one sample per AGU for each time point) and enabling the exploration of the full fungal biodiversity present in the samples, which is lower than the number of bacterial species. This approach allowed for cost and resource optimization without compromising statistical power. We hope that the Reviewer will agree with this decision.

- In all 8s, the scientific names must be written in italics.

As per Reviewer's suggestion, we modified all scientific names of bacterial genera in all the figures, when necessary (Figures 3, 5, 6 and 8, and Supplementary Figures 3 and 4).

- L. 158-160: this statement is not clear. The authors should clearly indicate whether the additional variables (rootstocks, agronomical practices, soil composition, etc.) are statistically significant factors that discriminate the sample groups.

We agree with the Reviewer that confounding factors between the different vineyard should be acknowledged and taken into account when discussing terroir microbiome geographical segregation. This is why we applied a permutational test to determine the spatial distances determined by the dissimilarity of rhizospheric microbial communities in different vineyards, also taking into account the aforementioned confounding factors (seasonality, agronomical practices and management, vine clone type, rootstock family, altitude, and soil composition). We included this information in Figure 4 description, and we also slightly modify the text in the Results section, (lines 163-164). We hope that this will increase clarity on the confounding factors, addressing the Reviewer's concerns.

- L. 200: remove one dot.

Modified as suggested.

- L. 362-364: the sampling strategy and the experimental design must be better described. It is not stated how many samples were taken per each site, the number of replicates, how each sample was obtained (how many plants/soil cores contributed to one sample), and so on.

We apologize with the reviewer for the lack of clarity. We modified (lines 407-408) to make the description of samples collection clearer. In particular, 6 rhizospheric samples and 1 bulk soil were retrieved for each of the 14 vineyards (84 rhizosphere and 14 soil samples) at each timepoint (4 in total), resulting in 336 root samples and 56 soil samples. As for the samples collection, the practical aspects are described in (lines 413-416). We hope that this new description will help clarify the sampling strategy.

- L. 417: how did you select these 14 subsamples?

In order to ensure that the selected subsamples were representative of their respective groups, the centroid of each group in the PCoA was calculated, and the sample closer to the centroid was selected. We hope that this will clarify our strategy to the Reviewer.

- L. 438: to my knowledge, the VSEARCH tool is used to detect chimeras, not to assign taxonomy. We applied QIIME2 pipeline that is a widely used software for amplicon sequencing data analysis. In QIIME2 pipeline, VSEARCH is one of the tools to be applied for taxonomic assignment, as described in Bokulich et al. (2018). Further information can be also found in the QIIME2 website.

L. 441: what about mitochondrial and chloroplast sequences? Didn't you consider them?

We excluded mitochondrial and chloroplast sequences from the analysis since they belong to the eukaryotic fraction of the sequenced DNA. As per Reviewer suggestion, we specified this in (line 436).

Reviewer #3 (Remarks to the Author):

In this study, the authors define the microbial terroir of vineyards in the "Consorzio del Vino Nobile di Montepulciano DOCG" PDO area in Italy. Using rhizospheres and soil samples, the study reveals microbiomes are "AGU-specific" in terms of taxonomic abundances and plant growth-promoting functions. The study emphasizes the importance of protecting microbial terroir and biodiversity for high-quality traditional wine production. In general, the result is interesting and important. However, more analysis needs to be done to strengthen the MS. A thorough review of the manuscript revealed the following observations.

1. The author is encouraged to highlight the novelty of this study, particularly in the context of existing literature on wine terroir. The integration of microbial composition and functional profiling with a longitudinal study and comprehensive metadata represents a unique and innovative approach.

We are truly grateful to the reviewer for their positive feedback. In an effort to contextualize the study more effectively, we have revised the introductory paragraph to highlight the study's unique contribution to the existing literature on wine-related microbial terroir (lines 71-74 and 86-89).

2. Line 385 - Consider clarifying if "ITS analysis" refers to fungal community analysis.

As per Reviewer's suggestion, we modified ITS with fungal (line 429).

3. Line 440 - Has the author conducted an analysis of the archaeal community? Given that the V3-V4 primer may amplify archaeal fragments, it is recommended to consider using the term "prokaryotes" instead of "bacteria" throughout the manuscript if the archaeal group is not excluded. Although some unspecific Archaeal amplification might occur when using the primers D-Bact-0341 and D-Bact-0785, they have been described by Klindworth et al. as one of the best primers pairs for the amplification of the domain Bacteria, with an estimation of Archaeal amplification of only 0.5%. We thus believe that the Archaeal presence on the described bacterial communities can be considered negligible. We hope that the Reviewer will agree with this consideration.

4. Line 462 - Please elaborate on the methods used to remove low-quality reads and reads originating from the host.

We used KneadData with default parameters that include the quality evaluation and removal of very repeated sequences by fastqc, the trimming/filtering of low-quality sequences (q<20) by Trimmomatic and the filtering of host-sequences aligning with the *Vitis vinifera* reference genome (RefSeq id: GCF_000003745.3). We included this information in lines 523-526.

5. Line 431 - Has the data been normalized prior to conducting alpha and beta diversity analyses? Yes, data was normalized to the lowest number of reads for all samples. In other words, the same number of reads were taken from each sample for conducting alpha and beta diversity analysis.

We included this information in the text in lines 488-489.

6. Line 465 - How were redundant genomes handled, particularly in instances where multiple metagenome-assembled genomes (MAGs) may be identical? Were non-redundant MAGs created for further investigation into MAG function?

The Reviewer is correct in pointing out that we could have been clearer in the description of this step in our approach. We have since revised our methodology to address this issue. Specifically, we have applied the BIN_REFINEMENT module of the MetaWRAP suite (Uritskiy et al., 2018), which allows for the pooling of redundant MAGs, as a potential solution of this issue (Lines 523-526).

7. In the Results section, consider removing repetitive information that has already been discussed in the Materials and Methods section (e.g., lines 188 and 224).

In order to avoid redundancy, we shortened the description of the selected PGP functions in the results section (lines 196-199). However, we did not eliminate it completely for clarity purposes. We hope the Reviewer will agree with this choice.

8. The interpretation of results at lines 228 and 300 may benefit from clearer implications. Consider exploring whether taxa with plant growth promotion traits were enriched in the rhizosphere due to interactions with host plants.

We are grateful to the Reviewer for this valuable suggestion. We conducted a comparative analysis of the PGP function levels between bulk soil and rhizospheres within the same AGU. Our findings revealed a strong correlation ($r > 0.98$, $p\text{-value} < 0.0001$, Pearson's correlation), indicating that a PGP function with a higher prevalence in the rhizosphere of an AGU was also more prevalent in the soil of the vineyard (lines 207-210).

9. Is the absence of certain genes attributed to their true absence or incomplete genome assemblies?

The Reviewer raises a valid point. It is our contention that in order to be considered for an application, a more profound investigation of the metagenomic data is required. This should integrate the metagenomic analysis with the isolation of microbial strains directly from the vineyards and subsequent study of their genomes, as well as the evaluation through specific functional tests. This additional consideration has been incorporated at the conclusion of the manuscript (lines 380-383). However, we wish to mention that in our manuscript the direct read mapping against a database of known PGP genes was also carried out, please line 201-217. These data do not suffer from biased due to the incomplete genome assembly and provide important glimpses on the different distribution of PGP functions in the Montepulciano territory, with the specific declinations at the AGUs level.

10. The unique functional profiles of microbial plant growth-promoting traits observed in the rhizosphere microbiomes of different agricultural units (AGUs) should be further linked to environmental factors such as the presence of phosphorus. Consider investigating if the enrichment of bacteria with phosphorus-solubilizing capabilities is influenced by varying phosphorus availability in different regions.

The Reviewer is right. The outcome regarding the microbiome's capacity for P solubilization aligns with the concentration of P in the soil. The S. Biagio AGU (western region of the territory) exhibits the highest soil P concentration and low potential for P solubilization, while the Argiano AGU (southeastern region) demonstrates the opposite trend. This suggests that a root microbiome with a heightened propensity for P solubilization may offset its deficiency in the soil (Supplementary Table 2). A short comment can be found in the Discussion section (lines 374-375 and 378-382).

11. Figure 2 shows that soil microbial communities from different countries do not strictly follow pedoclimatic zones. Investigate the potential reasons for the observed clustering in Figure 1.

The different countries included in Figure 2 can include different pedoclimatic regions, and this would explain why samples from the same country could have a wide distribution. However, the variable "country" allowed us to group the samples in the PCoA bidimensional space with a significant segregation between groups, both at a "country" level and within the Italian samples, considering different regions. We hope that this explanation will be sufficient to address the Reviewer's concerns.

12. Line 289 - The absence of a sampling season effect on grapevine rhizosphere communities is intriguing. Explore further whether the grapevine maintains a stable rhizosphere community regardless of seasonal fluctuations.

We apologize for the misunderstanding on this part. We did not investigate seasonal fluctuation of the grapevine rhizospheric community because we found that the geographical segregation of the AGUs microbiome was robust to seasonality (Procrustes test, $p\text{-value} \leq 0.01$). This does not mean that fluctuations did not occur at all, however we believe that the main factor driving microbiome

differentiation was geographical origin at a very local scale, rather than plant maturity and season. We described this test in lines 163-164 and this is why we decided not to discuss longitudinal analysis at specific timepoints. We hope that the Reviewer agrees with our choice.

13. Additional considerations and revisions may be needed based on the above feedback for enhanced clarity and depth in the manuscript.

We hope that our answers to the Reviewer's comments above will be sufficient to enhance the manuscript clarity.

14. This reviewer is missing information regarding the diversity of the microbial community? Are there differences in bacterial diversity between the sites? Why this part is not done. It would be interesting to investigate the microbial richness and diversity between the locations.

We thank the Reviewer for giving us the opportunity to implement this point. We initially did not include this information because we did not consider it relevant for the site description, since we observed significant but not geographically related alpha-diversity differences between AGUs. However, for completeness' sake, we followed the Reviewer's suggestion and included the alpha-diversity representation for both bacterial and fungal communities in the new Supplementary Figure 2. We included this new information in the text in lines 169-176.

15. The authors have a very nice data that could be explored more – for instance, is there any association between the bacterial and fungal community?

We have conducted this type of analysis and found a correlation between the fungal and bacterial communities (Procrustes test, p -value = 0.03). However, we believe that this detail may not be relevant for the purposes of the paper's narrative. If the reviewer agreed, we would not include it within the manuscript.